



# Measurements of precipitation in Dumont d'Urville, Terre Adélie, East Antarctica

Jacopo Grazioli[1], Christophe Genthon[2], Brice Boudevillain[3], Claudio Duran-Alarcon[3], Massimo Del Guasta[3], Jean-Baptiste Madeleine[4,5], and Alexis Berne[1]

[1]Environmental Remote Sensing Laboratory (LTE), École Polytechnique Fédérale de Lausanne (EPFL), Lausanne, Switzerland
[2]Univ. Grenoble Alpes, CNRS, IGE, F-38000 Grenoble, France
[3]Istituto nazionale di Ottica, INO-CNR, Italy
[4]CNRS, UMR 8539, Laboratoire de Météorologie Dynamique (LMD), IPSL Climate Modeling Center, Paris, France
[5]Sorbonne Universités, UPMC Univ Paris 06, UMR 8539, Laboratoire de Météorologie Dynamique (IPSL), Paris, France

*Correspondence to:* Alexis Berne (alexis.berne@epfl.ch)

**Abstract.** The first results of a campaign of intensive observation of precipitation in Dumont d'Urville, Antarctica, are presented. Several instruments collected data from October 2015, including a polarimetric weather radar (MXPol), a Micro Rain Radar (MRR), a weighing gauge (Pluvio$^2$), and a Multi-Angle Snowflake Camera (MASC). These instruments collected the first model-free measurements of precipitation in the region of Terre Adélie (Adélie Land), including of precipitation micro-
physics. Microphysical observations during the austral summer 2015/2016 showed that, close to ground level, aggregates are the dominant hydrometeor type, together with small ice particles (mostly originating from blowing snow), and that riming often occurs. Contamination of the Pluvio$^2$ measurements in windy conditions is observed and partly removed through synergistic use of MRR data. The yearly accumulated precipitation of snow (300 m above ground), obtained by means of a local conversion relation of MRR data, trained on the Pluvio$^2$ measurement of the summer period, is estimated to be 815 mm of water equivalent,
with a confidence interval ranging between 739.5 to 989 mm. Climatological data obtained from satellite-borne radars, and the ERA-Interim reanalysis of the European Center for Medium -Range Weather Forecasts (ECMWF) both provide lower yearly totals: 655 mm for ERA-Interim, while 679 mm for the climatological data over DDU. ERA-Interim seems to overestimate the occurrence of low-intensity precipitation events especially in summer, while visual observations conducted at the research stations all year long seem to underestimate it. Overall, this manuscript provides insightful examples of the added values of
precipitation monitoring in Antarctica with a synergistic use of in-situ and remote sensing measurements.

## 1   Introduction

The ice sheets of Antarctica contain about 90% of the world's ice and thus its evolution has potential impacts at a global scale. It conditions the evolution of the sea level height (Rignot et al., 2011; DeConto and Pollard, 2016), and the radiative budget of the lower atmosphere. In this context, the quantification and prediction of the surface mass balance (SMB) of the Antarctic
ice cap is a pressing scientific topic of investigation in order to understand if the continent is losing or gaining ice, and at what rate (Vaughan et al., 1999; Lenaerts et al., 2016).



Precipitation is an important component of the SMB as it represents, together with vapour deposition, the only net input of water and ice at the continental scale (Lawson and Gettelman, 2014). Precipitation is unfortunately also very difficult to monitor at high latitudes. The major problems hampering classical measurement techniques in Antarctica are: in the interior, the sparsity of human installations over a very large area, the extremely low temperatures and low precipitation intensities, and on the coasts the very strong katabatic winds blowing from the interior. Additionally, the complex logistics of Antarctic installations causes further difficulties and limitations for measurements to be conducted.

Until recently, information about precipitation that could be provided on the continent was obtained indirectly by analysing moisture transports, glaciological surface-based observations (Bromwich, 1990) and reanalysis of numerical weather prediction models (Bromwich et al., 2011). Additionally, long-running but qualitative human observation records of clouds and precipitation have been collected in some scientific stations with staff dedicated to meteorological measurements (König-Langlo et al., 1998). Recent research proposed a first model-free climatology of precipitation over a large part of the continent (Palerme et al., 2014, 2016) by exploiting the potential of the profiling radar on-board the CloudSat satellite, which is able to sample large horizontal areas but limited by the inability to measure precipitation at altitudes below a so-called "blind-range" above ground (1200 m above the surface for CloudSat).

In order to validate and to improve the performance of the models, and to constrain satellite-based measurements, it is necessary to establish and maintain for the medium to long term some in-situ observation sites, instrumented with precipitation measurement devices as autonomous and accurate as possible. There is therefore the need, before anything else, for accurate measurements of precipitation, including at the very local scale (Frezzotti et al., 2004; Schlosser et al., 2010; Welker et al., 2014). A recent effort in this direction was the establishment of an observatory in the escarpment zone of Dronning Maud Land, East Antarctica (Gorodetskaya et al., 2015). The synergy of in-situ and remote-sensing measurements allowed very first statistics of cloud and precipitation (Gorodetskaya et al., 2014, 2015) which showed that a few intense precipitation events govern the SMB in the area; measurement combinations have also been used to evaluate the quality of satellite-based precipitation products (Maahn et al., 2014) provided by CloudSat. It has been shown that the blind range of CloudSat, in the area of the measurements, can lead to an underestimation of precipitation amount on the order of 10% and an underestimation of the occurrence frequency on the order of 5%. The installation in DML can be considered the first well documented observatory in Antarctica that included precipitation measurements from remote sensing and in-situ instruments. An earlier effort involved co-located measurements of precipitation using radar and precipitation gauge, and was conducted at the Showa[1] Japanese station (Konishi et al., 1998), but very limited information about the outcome of those measurements is yet available in the literature.

In this work we present the results of an intensive observation campaign during the austral summer 2015-2016 and a first year of precipitation measurements conducted in the French base Dumont d'Urville, Terre Adélie, from November 2015, until November 2016 (and still ongoing). The data were collected in the framework of the *APRES3* project (Antarctic Precipitation, Remote Sensing from Surface and Space, see http://apres3.osug.fr). We provide statistics of precipitation quantity and occurrence, and we compare them with model reanalyses and with the visual observations collected by the French meteorological

---

[1]Sometimes spelled "Syowa"



office (Météo France) all year long. The goals of the manuscript are to contribute to a better quantification of precipitation in Antarctica (also by evaluating the products of numerical weather models) and to underline the innovative and promising aspects of the data collected until now, that may serve as an example for long-term monitoring of precipitation in other Antarctic regions. The paper is structured as follows: Sec. 2 describes the precipitation observation methods employed and inter-compared,

Sec. 3 lists the most relevant results, which are discussed and put into perspectives in Sec.4. Section 5 provides the summary and conclusions of the paper.

## 2    Methods

We present here data collected in a coastal location of Antarctica: the station Dumont d'Urville (hereafter DDU). The base is situated in Terre Adélie, $-66.6628$S, $140.0014$E, (41 m above sea level), on a coastal location highlighted in Fig. 1. This region

is located at the transition between the Antarctic continent and the Southern Ocean, where the terrain, which slopes downward from the inner continent to the coast, meets the ocean.

### 2.1    Climate and operational measurements

The climate at DDU is relatively mild in terms of temperatures, with minima rarely below -30°C, and maxima above 0°C in January and December, as illustrated in Fig. 1 (b). On the contrary, the wind regime is more extreme: in the low layer of

the atmosphere the dominant winds are katabatic coming from the inner continent, and the dominant wind origins are always between 90° (East) and 180° (South), as illustrated in the wind rose of Fig. 1 (b). Because of the intensity and persistence of the winds, which are able to reach hurricane force, Terre Adélie has been often described as the windiest place on planet Earth (e.g., Wendler et al., 1997). Standard measurements of atmospheric variables (temperature, wind, humidity, pressure, to name a few) are collected regularly all year long by the French meteorological service (Météo France), and a balloon radiosounding is

launched daily at 00UTC. Balloon soundings have been regularly conducted since 1956 at DDU. Visual observations of cloud, precipitation, and present weather are recorded as well. It is worth noting that at this location the visual observations are the only daily in-situ archive of past precipitation occurrence in DDU.

### 2.2    APRES3 Instruments

Several instruments were deployed at DDU, starting from November 2015. The instruments were as illustrated in Fig. 2, and

they are listed in Table 1. A Micro Rain Radar (MRR hereafter) was installed within an existing radome and has collected measurements uninterrupted since the 22 November 2015. This radar system is used to vertically profile clouds and precipitations with a resolution of 100 m at height levels ranging from 341 m to 3141 m above sea level[2]. During the measurement period, a resolution of 100 m was set for the instrument, and processed data was collected with a temporal resolution of one minute. The potential of the MRR to monitor polar regions has already been highlighted by the works of Gorodetskaya et al. (2015)

and Maahn et al. (2014). The simplicity of its deployment and operation makes it an attractive tool for long-term measurements

---

[2]300 m is the 3$^{\text{rd}}$ range gate of the MRR, where the first valid measurements are available, and 41 m is the altitude of DDU.





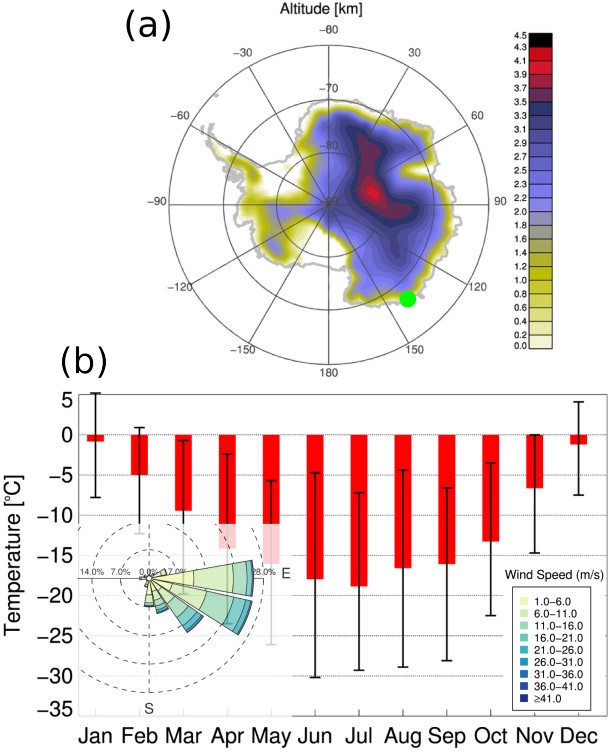

**Figure 1.** (a) Map of Antarctica with digital elevation model (domain south of 60°S). A green filled circle locates the station Dumont d'Urville. (b) Temperature statistics in Dumont d'Urville, based on data collected at 1 minute time resolution in the period 2011-2015. The red bars locate the mean value and the black error bars are used to highlight the 1% and 99% quantiles. Overlayed: Wind rose (origin and intensity) statistics.

in places with complex logistics and with limited possibility of support, in the case of instrumental failures. The raw K-band reflectivity measurements collected by the MRR were first processed with the method proposed by Maahn and Kollias (2012), then converted to X-band reflectivities and in a third step to snowfall intensities. Additional information about the processing of the MRR data is provided in Sec. 2.2.1.

A second radar, named MXPol (Mobile X-band dual-Polarization) collected measurements in the months of December 2015 and January 2016. This system, described in Schneebeli et al. (2013) and in Scipion et al. (2013), is a scanning dual-polarization Doppler radar. During its operation period at DDU, it was mainly collecting data at 75 m radial resolution and a maximum radial distance of 30 km, mostly conducting different types of scans within a repeating scanning sequence of 5 minutes: (i) Plan Position Indicator (PPI) scans, i.e. quasi-horizontal slices of the atmosphere, (ii) Range Height Indicator

(RHI) scans, i.e. vertical slices of the atmosphere, and (iii) static profiles, as the ones performed by the MRR.

A depolarization lidar (e.g. Del Guasta et al., 1993), deployed at a distance of about 200 m from MXPol, collected data in December 2015 and January 2016, as a test-bed for future long-term installation of a similar device. Lidar measurements allow



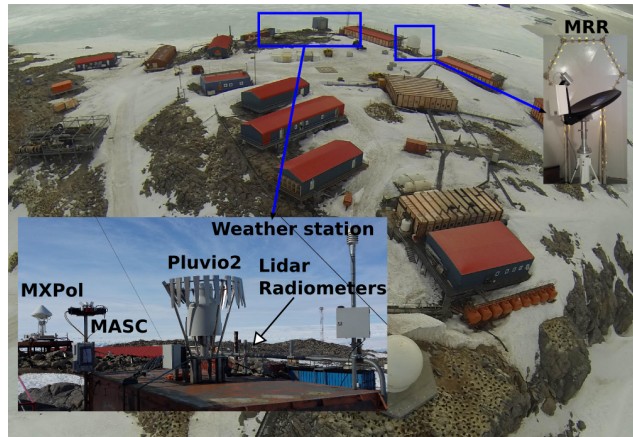

**Figure 2.** Main instruments deployed at DDU over the time period ranging from November 2015 to November 2016, and some of them still deployed.

for discrimination of the phase of the tropospheric clouds and detection of the occurrence of supercooled liquid water, and they complement the observations of ground-based weather radars, that are often not sensitive to this type of cloud particles.

An example is given in Fig. 3, where the timeseries of MRR reflectivity, lidar signal and depolarization ratio are shown for the day of the 15[th] December 2015. Supercooled liquid water appears in the lidar data as a layer of enhanced signal and low depolarization ratio (e.g. Del Guasta et al., 1993; Hogan et al., 2003), often when no MRR signal is visible. On the contrary, when precipitation occurs, (around 04 UTC, and from 14 to 24 UTC) the lidar signal gets fully attenuated in the lowest 500 m while the MRR is still able to sample the vertical precipitation column.

A weighing precipitation gauge (Pluvio$^2$, manufactured by OTT) was deployed from November 2015 to January 2016. This instrument provides the liquid water equivalent of snowfall falling within its measurement area at a time resolution of one minute. To avoid excessive contamination of precipitation signals by blowing snow, the Pluvio$^2$ was installed at a height of about 3 m above ground and its inlet was protected with a standard wind fence designed by the same manufacturer as the instrument. It must be noted that this wind shield is not sufficient to avoid the adverse effect of strong wind (frequently

occurring at DDU).

Located close to the weighing gauge, a multi-angle snowflake camera (MASC hereafter) was deployed, also in the period from November 2015 to January 2016. This instrument collects high-resolution photographs of snowflakes in free fall, while they cross its sampling area (Garrett et al., 2012), thus providing information about snowfall microphysics. To complete the set of in-situ measurements, a weather station was installed close to the Pluvio$^2$ and the MASC, to sample the environmental

conditions in the close proximity of their measurements.





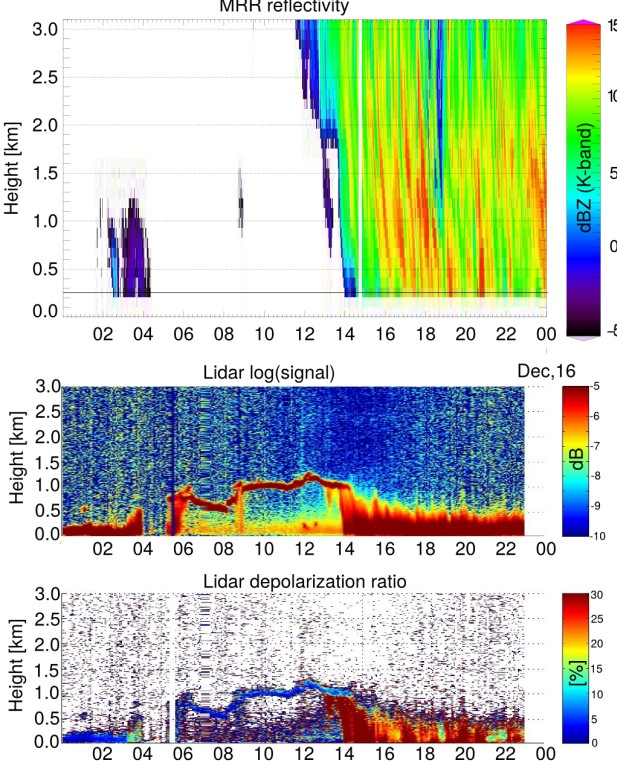

**Figure 3.** Example of a timeseries (time-height image) of MRR data and lidar data for the 15[th] December 2015.

**Table 1.** Non-exhaustive list of the instruments deployed at DDU in the framework of APRES3. Only the APRES3 instruments with a certain relevance for precipitation monitoring are listed here.

| Name | Deployment period | Instrument type | Measurement | Reference |
|---|---|---|---|---|
| MRR | 2015.11.21 - ongoing | FMCW[a] radar profiler, 24 GHz | Clouds / Precipitation | Maahn and Kollias (2012) |
| MXPol | 2015.12.07 - 2016.01.31 | Dual-pol Doppler radar, 9.41 GHz | Clouds / Precipitation | Schneebeli et al. (2013) |
| Lidar | 2015.12.15 - 2016.01.29 | Depolarization lidar | Clouds / Precipitation | Del Guasta et al. (1993) |
| MASC | 2015.11.11 - 2016.01.31 | Snowflake imager | Precipitation / Blowing Snow | Garrett et al. (2012) |
| Pluvio[2] | 2015.11.17 - 2016.01.31 | Weighing gauge | Precipitation | Colli et al. (2014) |
| Biral VPF-730[b] | 2015.12.03 - 2015.12.25 | Present weather sensor | Visibility / Present weather | – |
| Weather station[c] | 2015.11.11 - 2016.01.31 | Weather station | T, RH, Wind | – |

[a]: Frequency Modulation Continuous Wave.

[b]: For the rest of the time this instrument was (and is) deployed on the Antarctic continent, about 5 km away from DDU.

[c]: The co-located weather station of Météo France is providing data all year long, uninterruptedly.



### 2.2.1 Pre-processing of MRR radar data

The MRR radar was co-located with MXPol for the period of the summer campaign 2015/2016 (See Table 1). The purpose of the former instrument at DDU is long-term monitoring, which involves exposure to the extremely windy winter conditions. It was decided, in order to avoid failures during the winter when no member of the scientific team is on-site, to install the MRR inside an existing radome previously used in the base for satellite communications, as shown in Fig. 2. Although this installation ensures protection and easy access to the instrument, it adds an unknown amount of attenuation to the measurements. For this reason the co-located MXPol measurements collected during the summer period are used to map the radome-affected reflectivity data provided at K-band (MRR) into $X$ band reflectivities.

The scatter plot in Figure 4 shows the comparison of reflectivity values measured by the MRR and by MXPol for data collected during the period of co-location of the instruments. Because overall the relation between the two sets of measurements is close to linear ($\rho^2 \approx 0.88$), and almost equivalent to a simple offset subtraction, the following conversion has been applied to MRR data:

$$Z_X = 0.99\, Z_K + 6.14 \pm \epsilon \tag{1}$$

where $Z_X$ ($Z_K$) [dBZ] is used to indicate reflectivity at $X$ ($K$) band, and $\epsilon$ is the measure of uncertainty of the linear relation with respect to the scatterplot of Fig. 4 (whose standard deviation of the residuals is $1.9\,\mathrm{dBZ}$). It is worth mentioning that $Z_K$ is originally obtained with the method of Maahn and Kollias (2012), who proposed an improved and innovative processing chain for MRR data collected in snow. Once mapped to X-band, reflectivity can be converted to snowfall rate $S$ rate by means of $Z$-$S$ power laws available in the literature. For example, the six relations proposed by Matrosov et al. (2009), and listed in Table 2 can be used. These relations were obtained by combining two different snowflake size distribution datasets, and three different mass-to-size relations. The error component of Eq. 1, and the large variability of the $Z$-$S$ relations, lead to very uncertain retrieval of snowfall rate. For this reason, we optimized a local relation, by fitting its two parameters in the $Z$-$S$ space given by the MRR and the Pluvio[2] measurements, during the Summer period 2015/2016. The local relation, also listed in Table 2, takes the form of $Z = 75.85\, S^{0.91}$. In order to mitigate the difference in sampling volume of the two instruments, it has been derived for hourly data. The 95% confidence intervals for the two parameter are 68.68-82.94 and 0.776-1.09 respectively.

### 2.2.2 Pre-processing of Pluvio[2] data

It has been observed that occasionally the values of equivalent water of the Pluvio[2] show a "phantom" accumulation (similar to that reported by World Meteorological Organization, 2014). In such cases, no precipitation was observed on site and no precipitation signal was visible in the MRR data but the content of the Pluvio[2] bucket was increasing. In order to censor these cases, we combined the information coming from remote sensing (MRR) and in-situ data (Pluvio[2]). More precisely, time steps when no signal was recorded by the MRR at its lowest available gate ($300\,\mathrm{m}$ above ground level), are considered "precipitation free" and any increase in the cumulative precipitation records of the Pluvio[2] is thus related to contaminations by blowing snow, vibrations, and other possible phantom accumulation processes. The assumption is that precipitation is extremely unlikely to





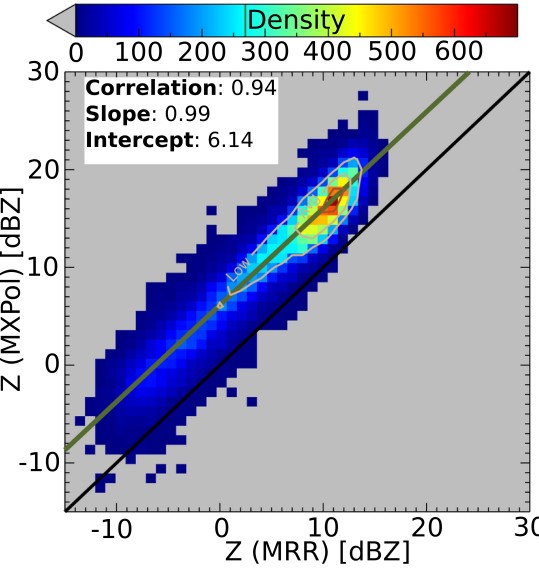

**Figure 4.** Scatter plot of reflectivity values at $9.41\,GHz$ (X-band, measured by MXPol) and at $24.3\,GHz$ ($K-$band, measured by the MRR) during the summer campaign 2015/2016.

**Table 2.** Parameters of the 6 conversion relations between radar reflectivity $Z$ and snowfall intensity $S$ [mmh$^{-1}$] of Matrosov et al. (2009), and the local relation, obtained using the instruments at DDU. In these relations, the radar reflectivity ($Z$) must be used in linear units [mm$^6$m$^{-3}$].

| Relation* | Equation |
|---|---|
| B90A (1) | $Z = 67\,S^{1.28}$ |
| B90B (2) | $Z = 114\,S^{1.39}$ |
| M2009 (3) | $Z = 136\,S^{1.30}$ |
| M2009 (4) | $Z = 28\,S^{1.44}$ |
| M2009 (5) | $Z = 36\,S^{1.56}$ |
| M2009 (6) | $Z = 48\,S^{1.45}$ |
| Local-DDU | $Z = 75.85\,S^{0.91}$ |

*: In parentheses, the way the relations were
numbered in Matrosov et al. (2009).

completely develop in the lowest $300\,m$ of the atmosphere. An example of the behaviour of this simple censoring filter can be found in Fig. 5 (a). From the end of October 2015 until the end of January 2016, about $14\,mm$ of liquid water equivalent snowfall have been removed, corresponding to about 21% of the uncensored data.

Figure 5 (b) shows the evolution of wind speed in the near proximity of the Pluvio$^2$ inlet and illustrates how the most intense phantom accumulations occur when the strongest wind peaks are observed, giving confidence in the proposed censoring



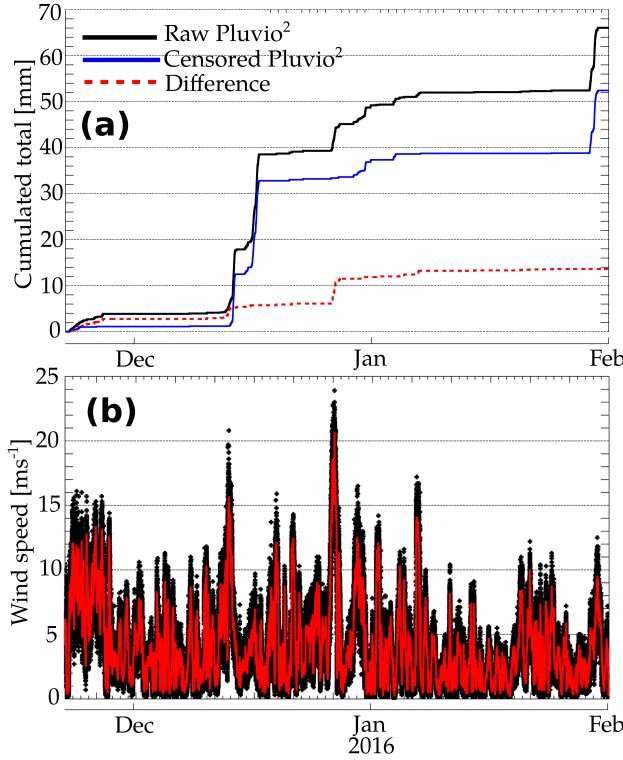

**Figure 5.** (a) Time series of Pluvio$^2$ untreated data, censored data (taking the MRR measurements as occurrence indicators), and their difference. (b) Wind speed measured in the near proximity ($\leq 2\,$m) of the Pluvio$^2$ inlet.

method. It must be noted that this pre-treatment cannot account for "mixed" cases (snow and blowing snow together), when the positive contribution of blowing snow and precipitation, and the negative contribution due to wind-induced loss of catching

100   efficiency occur together.

## 2.3   Additional data

Due to the lack of both short- and long-term precipitation measurements, net precipitation estimates in Antarctica have been obtained from numerical weather prediction models (e.g., Cullather et al., 1998; Schlosser et al., 2010). Among the available model-based products, the ERA-Interim global reanalysis provided by the European Center for Medium-Range Weather Forecasts (ECMWF) is considered as a reference as it is thought to provide the best representation of precipitation variability (Bromwich et al., 2011; Palerme et al., 2014), and the best agreement with satellite-borne measurements (Behrangi et al.,

5   2016; Palerme et al., 2016). For this reason, and because of its global coverage and easy access, ERA-Interim reanalysis is used here to illustrate the performance of model-based methods. The analyses at 00 UTC and 12 UTC, and forecast time steps of 6, and 12 h are used in the present work. The spatial resolution of ERA-Interim is 0.75° × 0.75°. To quantify precipitation, the model variable "tp" (total precipitation) is used here.



## 3 Results

### 3.1 Microphysical observations during Summer 2015/2016

The period between November 2015 and January 2016 was heavily instrumented with devices that are able to provide micro-physical information about precipitation, thus microphysical aspects are better documented during the summer months. While a complete investigation of the dominant microphysical processes and the small-scale dynamics of precipitation in this region is beyond the scope of this paper, it is worth to investigate the most immediate microphysical parameter: the hydrometeor type.

Hydrometeor types have been recorded at the ground level by the MASC instrument through classification of individual particle pictures with the recently developed method of Praz et al. (2017), able to classify individual hydrometeors into six classes (and melting snow), and to assign to them a continuous riming degree index ranging from 0 to 1, with 1 corresponding to fully developed graupel. The choice of the available classes is based on the widely used scheme of (Magono and Lee, 1966). The instrument being close to the ground level, both precipitation and blowing snow particles are recorded and classified.

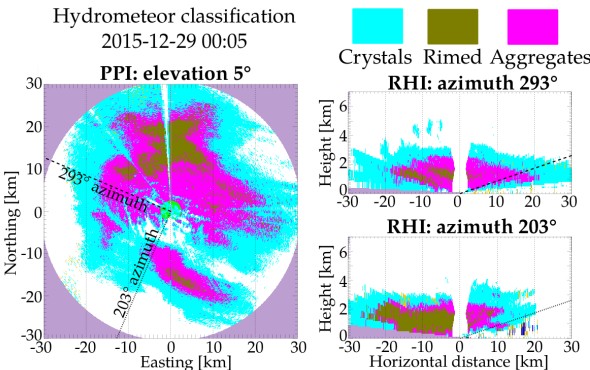

**Figure 6.** Example of a PPI and two RHI scans collected by MXPol on the 29[th] around 0014 UTC. The variable displayed in the image is the hydrometeor classification, obtained with the method of Grazioli et al. (2015).

A second classification method is obtained from the polarimetric data of MXPol, that can be converted into hydrometeor measurements with an hydrometeor classification algorithm (Grazioli et al., 2015). Despite the drawbacks of being an indirect method and not being able to retrieve at near-ground heights (because of ground clutter contamination in the radar data), it has the advantage of providing hydrometeor types over large domains and at different height levels, as shown in Figure 6, that illustrates PPI and RHI scans of the hydrometeor classification for a case where all its ice-phase hydrometeor classes are observed. This classification method discriminates pure snowfall into three categories: crystals, aggregates, and rimed particles. Figure 7 illustrates the statistical distribution of those three classes for the period of operation of MXPol, as a function of height. Below 2000 m, the proportion of the three hydrometeor types it is relatively constant, with about 10% of rimed snowflakes, 40% of aggregates and 50% of crystals. With increasing height and getting closer to the cloud top, aggregates and rimed snowfall rapidly disappear while crystals constitute the dominant hydrometeor class.





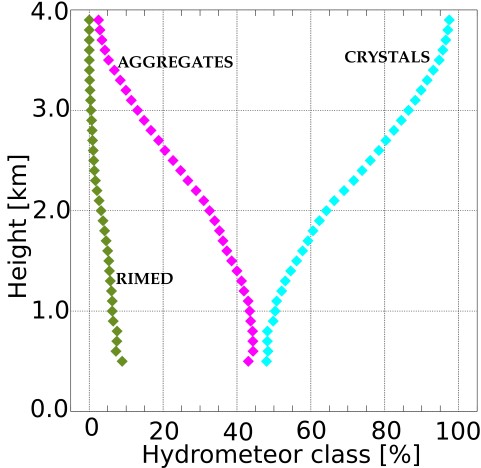

**Figure 7.** Fraction of occurrence of different hydrometeor types as a function of height, obtained over the period of operation of MXPol, with the hydrometeor classification algorithm of Grazioli et al. (2015).

The classification obtained with the MASC and the method of Praz et al. (2017) has been summarized in Figure 8. At ground level, the majority of the particles (54%) are classified as "small", indicating hydrometeors too small for their geometry and texture to be properly captured by the MASC. This proportion is three times higher than similar measurements collected in the Swiss Alps, while the proportion of the other hydrometeors is similar among the two different locations (not shown here). The occurrence of strong katabatic winds being a major difference between the sites, it can be assumed that the large majority of these "small particles" observed at DDU is associated with blowing snow. Also from this classification we observe that riming occurs, 11% of the particles are fully rimed (graupel), all the other hydrometeor types, have a riming degree ranging mostly from 0.1 to 0.5, and sometimes larger than 0.5 for the aggregates.

While the outcomes of the classification from MXPol and the MASC are not directly comparable because of the differences in measurement height, sampling volume, and available classes, it must be underlined that radar measurements are very sensitive to the size of the hydrometeors. Thus, a few large aggregates within a radar sampling volume, will dominate and overcome the signal coming from smaller hydrometeors. This can partially explain the different proportion of aggregates observed by MXPol (about 40% at a 400 m height), and by the MASC (93%). A second contribution to this difference may be low-level mechanical breakup of the aggregates (e.g. Vardiman, 1978). A third, and very likely, contribution is the contamination of blowing snow in the MASC measurements, namely in the "small" hydrometeor class. If, assuming that most of the "small" particles originate from blowing snow, they are removed from the statistics, then aggregates account for 41% of the hydrometeors, a value much closer to the 40% obtained with the classification of MXPol.





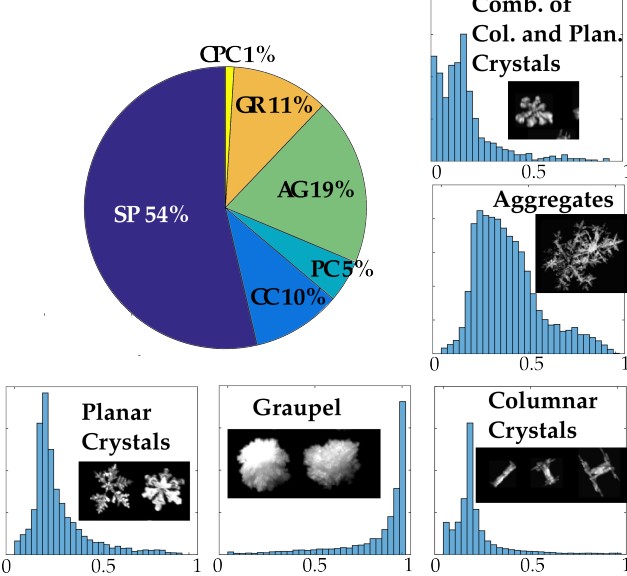

**Figure 8.** Pie-chart of the hydrometeor types classified by the MASC instrument in the period from the 21[st] November 2015 until the end of January 2016. The histograms shows the distribution of a riming index, ranging from 0 (unrimed) to 1 (fully rimed), for the particles of each hydrometeor class. The riming index is undefined for "small" particles.

## 3.2 One year of MRR precipitation data

The MRR instrument collected precipitation data uninterruptedly, covering the evolution of precipitation over the entire year. It offers therefore an interesting ground-based (but remotely sensed) set of data to compare with model-based data and with available human observations. Figure 9 shows the estimates coming from the MRR and other available sources of information over a year of measurements. The agreement of the local MRR relation with the Pluvio[2] is good over the summer period (Dec-January), during which the relation was obtained. In this period also the estimate of ERA-Interim provides a total cumulated precipitation within the uncertainty envelope of the optimized $Z$-$S$ relation, even though the curves do not co-fluctuate well. The optimized $Z$-$S$ relation provides estimates that are close to the B90A relation of Matrosov (2009).

The months with the highest cumulated precipitation were the late fall and winter months of May and June, and the month of September. Seasonally[3], Summer was the driest season, contributing only 11% of the yearly total, compared to values close to 30% for spring, 34% for fall, and 25% for winter (Table 3). The ERA-Interim totals of each month of the comparison period are within what could be observed in the period 1995-2015, with the exception of September that was the snowiest since 1995.

---

[3]We refer here to the seasons of the mid latitudes of the Southern Hemisphere. Summer: December, January, and February. Fall: March, April, May. Winter: Jun, July, and August. Spring: September, October, and November.



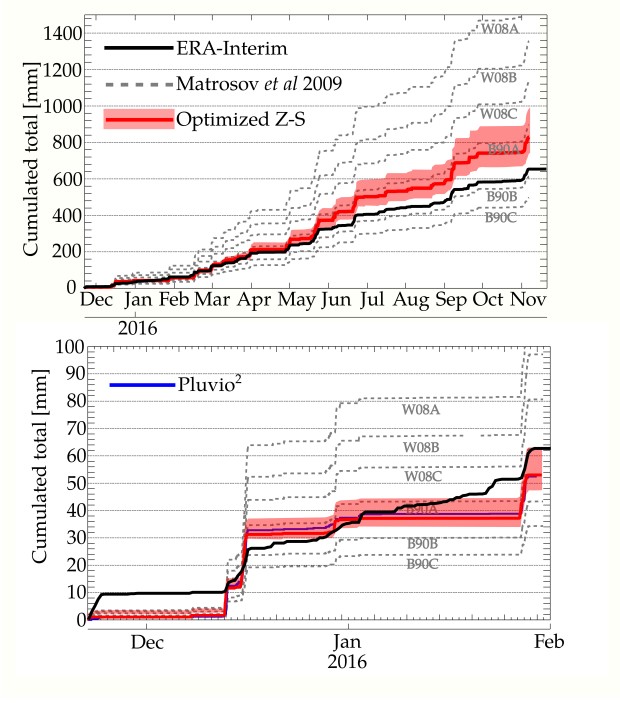

**Figure 9.** Cumulated snowfall liquid water equivalent from Pluvio[2] (when available), censored from phantom precipitation, MRR (see relations of Table 2, and ERA-Interim data. Top panel: full year of measurements. Bottom panel: focus on the Summer campaign 2015/2016.

### 3.3 Precipitation occurrence

Long term precipitation data records in Antarctic areas are often only visual observations of precipitation occurrence. For this reason, comparing precipitation occurrence measurements is a way to better understand the quality of this longer source of information. For the year 2015-2016, we can compare in terms of occurrence the information coming from ERA-Interim, Pluvio[2], MRR, and the visual observations archived by Météo France. We deal at first with occurrence at the daily scale, and we define it for the MRR and ERA-Interim as precipitation exceeding a given threshold over a given duration. A threshold of $0.07\,\mathrm{mm}$ over 6h was proposed by Palerme et al. (2014), and we thus take a value of $0.28\,\mathrm{mm\,d^{-1}}$ as a first guess. However, the choice of a unique threshold is delicate, and we apply also a minimum (maximum) threshold of $0.001\,\mathrm{mm\,d^{-1}}$ ($1\,\mathrm{mm\,d^{-1}}$) to cover any value that it appears reasonable to assume. Figure 10 shows the number of days with precipitation recorded during each month of the measurement period.

As a past reference, also shown is the historical record of precipitation occurrences from visual observations for the preceding years (1981 to 2015) in green with variability range. This allows us to observe how the year under investigation had an extremely dry January, and an extremely snowy September, while the other months are within the range of past occurrences. Overall, ERA-Interim mostly overestimates precipitation occurrence with respect to the MRR, especially in Summer, while the visual observations underestimate it. For January and December, when the Pluvio[2] was in operation, it is in agreement with





**Table 3.** Monthly cumulated precipitation of snow (liquid water equivalent) from the MRR, using the locally optimized $Z-S$ relation and the confidence interval of its parameters, and ERA-Interim data. The mean, minimum, and maximum snowfall of each month for ERA-Interim data from 1995 until 2015 are also shown.

| Month | $\mathbf{MRR}^{Min-Max}_{2015/2016}$ | $\mathbf{ERA}_{2015/2016}$ | $\mathbf{ERA}^{Mean}_{1995-2015}$ | $\mathbf{ERA}^{Min-Max}_{1995-2015}$ |
|---|---|---|---|---|
| January | 13.6 - 19.0 | 27.6 | 52.9 | 17.7 - 106.4 |
| February | 35.3 - 45.7 | 33.4 | 44.9 | 18.8 - 81.5 |
| March | 76.0 - 93.0 | 80.9 | 55.3 | 9.9 - 203.3 |
| April | 49.9 - 77.3 | 35.4 | 51.5 | 11.8 - 114.3 |
| May | 126.0 - 160.0 | 113.3 | 42.7 | 5.7 - 108.7 |
| June | 115.0 - 158.8 | 80.2 | 36.5 | 4.9 - 81.7 |
| July | 28.6 - 34.3 | 36.5 | 48.6 | 2.6 - 96.6 |
| August | 37.6 - 46.5 | 27.6 | 61.8 | 16.9 - 113.6 |
| September | 147.6 - 208.9 | 113.3 | 44.2 | 4.4 - 75.2 |
| October | 3.4 - 4.7 | 8.3 | 30.9 | 0.1 - 117.4 |
| November | 75.2 - 100.5 | 72.9 | 22.5 | 1.6 - 59.6 |
| December | 31.3 - 40.3 | 25.4 | 51.4 | 17.5 - 131.4 |
| **Total [mm]** | 739.5 - 989.0 | 654.8 | 543.1 | 392.8 - 702.5 |

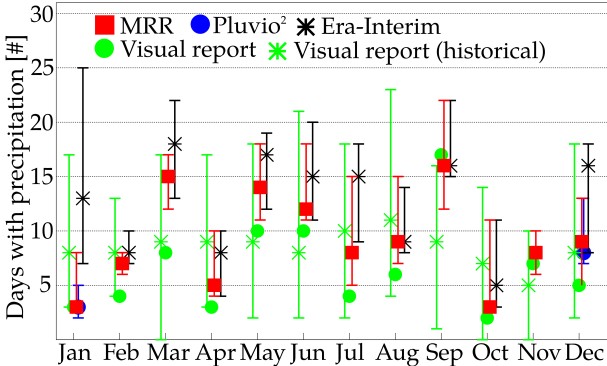

**Figure 10.** Precipitation occurrence at the daily scale. The error bars (where applicable) come from the use of a threshold of $0.001\,\mathrm{mm\,d^{-1}}$ (upper limit), and $1\,\mathrm{mm\,d^{-1}}$ (lower limit), while the central points are calculated with a threshold of $0.28\,\mathrm{mm\,d^{-1}}$ following the threshold of Palerme et al. (2014). The bars of the historical visual reports indicate instead the minimum and maximum occurrence in the period 1981-2015.

the MRR. Given the measurement correction principle based on false detection described in Sec. 2.2.2, this implies that no miss-detection is evident.



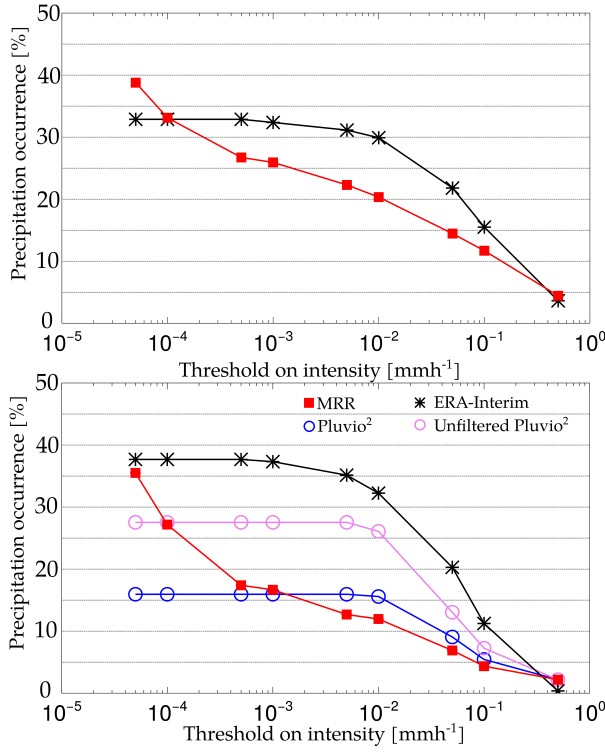

**Figure 11.** Precipitation occurrence as a function of the threshold on precipitation intensity for the full year (top), and summer period (bottom) respectively. The precipitation occurrence time scale is 6 h.

A good example of the overestimation of occurrence by ERA-Interim is shown in Fig. 9, bottom. The period between the $10^{th}$ and $25^{th}$ of January is seen as dry by the MRR and Pluvio[2], while several low-intensity precipitation events appear in the ERA-Interim time series. With the cumulated compensation resulting from the overestimated occurrence, at the end of January the total cumulated precipitation of ERA-interim gets close to the one of the Pluvio[2].

Figure 11 shows, at a 6 h time scale, the evolution of precipitation occurrence as a function of a given average precipitation intensity threshold, for the full year of observations (top panel) and for the summer campaign (bottom panel). Also here the overestimation of precipitation events by ERA-Interim is evident, especially in Summer period. The curves of Pluvio[2] and MRR overlay relatively well, with the exception of very low thresholds were the minimum intensity recordable by the Pluvio[2] becomes a limitation, due to the quantization effect. The black curve of ERA-Interim is above the red curve of the MRR for low-intensity precipitation thresholds, and below it for higher precipitation thresholds. The two curves first cross each other at the yearly scale where the threshold is approximately $10^{-4}\,\mathrm{mmh^{-1}}$, and then at $0.5\,\mathrm{mmh^{-1}}$.

Visual observations provided by by Météo France are not limited to precipitation, but several present weather codes are archived. At DDU, those are SYNOP codes belonging to the group 7wwW1W2. In the codes recurring at DDU, three types of phenomena are mostly documented: clouds (codes 1-3), blowing snow (codes 36-39), and snow (codes 22 and 71). We consider





**Table 4.** Contingency table between occurrence of near-ground (300 m) MRR signal (columns) and visual observations of snow and blowing snow conducted by Météo France (rows). A threshold on the MRR data of $10^{-2}\,\mathrm{mmh}^{-1}$ is used to discriminate between "dry sky" and "precipitation". The elements of the contingency table are normalized to the total number of observations, and sum up to 100%.

|  | Precip. (MRR) | Dry (MRR) |
|---|---|---|
| **Blowing snow** | 14 | 44.4 |
| **Snow** | 34.9 | 6.7 |

here, given the goal of the paper, the snow and blowing snow related codes and we disregard the observations of clouds. The observations are conducted several times during each day, and we compare them here with the MRR measurements. This is shown in Table 4, where the MRR observations at 300 m height are compared with the visual observations of snow and blowing
snow.

Given the intrinsic difference between those observations, it is not possible to take one as an overall reference in the confusion matrix. However, it can be assumed that visual observations are better at reporting blowing snow, because they are conducted at the ground level, while MRR measurements at 300 m above ground are better at reporting snowfall. The most interesting outcomes of this comparison are the following ones. First, there is a good correspondence between occurrence of
snow according to the MRR and visual observations of snow. Second, blowing snow occurrence is not well captured by the MRR. When visual observations report blowing snow, the MRR mostly does not report any occurrence. 44.4% of the visual reports analysed here correspond to cases where blowing snow has been observed at the ground level, but no valid signal has been recorded by the MRR. This comparison is to a certain degree dependent on the threshold used to discriminate "dry" and "precipitation" in the MRR data. However, similar results have been obtained for various threshold levels (not shown here).

**4    Discussion**

**4.1    Microphysical observations**

The microphysical observations, collected during the austral summer 2015-2016, and illustrated in Figs. 7 and 8 suggest that even at this location on the Antarctic coasts, riming is an important microphysical process. From radar retrievals, close to the ground level, about 10% of precipitation is rimed. According to the MASC classification, 11% of the hydrometeors are fully
rimed (graupel) and most of the other hydrometeor types have a degree of riming greater than 0, in particular aggregates that tend to be larger and easier targets for riming (e.g. Houze and Medina, 2005). The presence of riming indirectly shows that mixed-phase clouds are often occurring and that supercooled liquid water is available in the regions of precipitation formation. This has been documented in the past at this location by Del Guasta et al. (1993), and it could be observed also in the test data collected with the depolarization lidar (see Table 1, and Fig. 3) during the summer period.





At the ground level the large majority of hydrometeors recorded by the MASC were small particles of non-discernible habit and non-definable riming degree, with an occurrence three times higher than similar measurements conducted in Alpine locations. This is probably the signature of the significant contribution of blowing snow to the near-ground snow flux, that is particularly effective in recirculating small and light particles (e.g. Mann et al., 2000; Gordon et al., 2009), but it could, in a minor part, be the result of the fragmentation of aggregates in the low-level of the atmosphere, where strong katabatic winds

blow.

## 4.2   Blowing snow

Apart from the observations collected with the MASC, the contribution of blowing snow was visible in the Pluvio[2] measurements, despite the protection of the single wind shield. As illustrated in Fig. 5, periods with wind speeds exceeding roughly $15 \, \mathrm{ms}^{-1}$ at the proximity of the inlet of the instrument generate phantom accumulations of precipitation that are then removed

if the MRR does not receive any signal at the same time at its lowest gate. The lowest gate of the MRR ($300 \, \mathrm{m}$ in this case) is considered to be high enough to be above the height of any wind-blown snow layer (Gordon et al., 2009; Scarchilli et al., 2010; Palm et al., 2011), which rarely exceeds $200 \, \mathrm{m}$ of vertical development.

   This synergetic use of remote sensing and in-situ measurements has the potential to both overcome in part the main problem of in-situ measurements in coastal regions of Antarctica (i.e., blowing snow), and to estimate the contribution of blowing snow

itself to the accumulation. For this reason this combination of instruments should be proposed again, and maintained for at least a full year. The main limitation is that this treatment of blowing snow applies conceptually only to pure blowing snow occurrences, and cannot detect cases when blowing snow occurs together with precipitation. In the case of our measurements, the total cumulated precipitation of the Pluvio[2] in the summer period drops from $66 \, \mathrm{mm}$ to $52 \, \mathrm{mm}$, after the censoring is performed. The removed portion is then $21 \, \%$ of the total raw cumulated precipitation.

   Regarding blowing snow, the comparison between the occurrence of signal at the lowest MRR gate and the visual weather

reports conducted by Météo France, summarized in Table 4, suggests that the MRR is not able to capture the occurrence of blowing snow events. This suggests that the MRR alone in this configuration is not an appropriate instrument to monitor blowing snow. Lower range gate spacing should be employed (lower than $100 \, \mathrm{m}$ used in the measurements shown here) if blowing snow is of interest.

## 4.3   Precipitation quantification

The quantification of precipitation in the coastal regions of Antarctica remains a difficult task, affected by significant uncertainty. This study however provides some estimates that help to contextualize the information available until now. Fig. 9 and the summary in Table 3 show that MRR estimates of total cumulated precipitation at the yearly scale can diverge significantly (from 484 to 1581 mm) if a range of standard $Z$-$S$ relations is used, while the use of a local $Z$-$S$ relation allows for a significant reduction in this range of values (from 740 to 989 mm).

ERA-Interim provides a yearly estimate of 655 mm for the measurement period from November 2015 to November 2016, about 10% lower than the lowest estimate obtained with the local $Z$-$S$ relation. It must be underlined once more that the



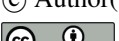

estimates provided by the MRR at DDU correspond to a minimum height of 300 (±50 m above ground corresponding to a range gate spacing of 100 m). This, together with the large grid of ERA-Interim, may contribute to the differences observed with the MRR. Interestingly, ERA-Interim initially is in agreement with the MRR precipitation values until March 2016, while

later on it underestimates them. This may also be due to a seasonal change in snowfall type, no longer representative of the summer snowfall events used to build the $Z$-$S$ relation.

As an external reference, the mean climatological estimate proposed by Palerme et al. (2014) is of 679 mm over DDU (climatology obtained for the period 2006 to 2011), a value not very far from the 2015/2016 measurements.

The year of measurements (2015/2016) was characterized by a significant inter-seasonal and inter-month variability. How-

ever, according to ERA-Interim records, the monthly totals are within what has been observed since 1995, with the exception of the snowy month of september.

### 4.4 Precipitation occurrence

Occurrence is an interesting parameter, because it is the only precipitation-related measurement that has been collected on the DDU base for a long time. In terms of precipitation occurrence we take as a reference the MRR, because visual reports are

discontinuous and affected by the limitations of visibility that can occur near the ground. Figure 11 shows that the year under investigation had some peculiarities: the month of september had the highest occurrence of precipitation since 1981, while january, february, and april all equalled the records of lowest monthly occurrence for the same period.

ERA-Interim generally overestimates the occurrence of precipitation, which could be caused by a sampling effect due to the much larger grid size of ERA-Interim with respect to the local MRR measurements, despite temporal integration to 6 hours

to reduce this effect. This overestimation is evident mostly in summer, in particular in december and january and it is well depicted by the timeseries of Fig. 9, bottom. With respect to the MRR, ERA-Interim tends to overestimate the occurrence of low-intensity events and underestimate the occurrence of high-intensity events (illustrated in Fig. 11), and an optimal threshold to match the two occurrences over the year of measurement (at 6h scale) is between 0.1 and 0.5 mm h$^{-1}$. Because the events of highest intensity did not occur in summer, this can contribute to explain the underestimation of ERA-Interim starting in March

5    2016.

Visual observations tend to underestimate the occurrence of precipitation, as shown in Fig. 11, but they rarely produce false alarms of precipitation, as visible in Table 4. In other words, when visual observations report precipitation, they are overall correct, but they can miss some occurrences, probably due to visibility limitations, human errors, confusion with wind-blown snow, and due to the discontinuous nature of human observations. In Fig. 11, we can observe no clear seasonality in the

underestimation of occurrence. For example a large underestimation is observed both in march and in july, while in june, april, and september the occurrence is very close between MRR and visual reports. This is to a certain extent surprising, as a larger missed detection rate could be expected during the dark winter months, when the reduced visibility may affect the human observations.



## 5   Summary and conclusions

In this paper we present unprecedented observations of precipitation collected at a coastal location of East Antarctica since October 2015. Several remote-sensing and in-situ instruments collected measurements during summer 2015/2016, and one (the MRR) has been operating continuously since then. These instruments have provided an insightful example of their usefulness to monitor precipitation on the Antarctic continent. It has been shown that radar data can be used to remove phantom accumulations from in-situ weighing gauge observations. These accumulations, occurring in high-wind conditions, and thus

assumed to be due to blowing snow and vibrations, accounted for 21% of the total cumulated precipitation of the summer period. Microphysical observations at the ground level, collected by the MASC in summer, showed that the large majority of hydrometeors (54%) were small ice particles of non-defined habit probably resulting from blowing snow, followed by aggregates (13%), and other hydrometeor types. Both from radar-based hydrometeor classification, and from MASC measurements, it appeared that riming is an active process. About 10% of the radar measurements at low-level were classified as containing

rimed hydrometeors, 11% of the hydrometeors were classified as fully developed graupel, and most of the other hydrometeors classified with the MASC showed riming degrees even larger than 0.5. The presence of supercooled liquid water, a necessary ingredient for riming, has been reported at DDU by previous studies and was evident in the lidar measurements collected in 2015.

One year of MRR data allowed for the estimation of the total yearly precipitation, from October 2015 until October 2016,

giving values ranging between 740 to 989 mm, at least 10% larger than that provided by ERA-Interim reanalysis (655 mm). The MRR estimates were based on a local reflectivity-to-snowfall rate, obtained on summer snowfall data only. Precipitation occurrence was generally overestimated by ERA-Interim with respect to the MRR, especially in the summer period, and was underestimated by the visual reports collected by Météo France. It is difficult to understand the reasons behind the overestimation of occurrence by ERA-Interim, which could be due to its microphysical parametrization or to a spatial resolution very

different from the one of the point measurements used as a reference. On the contrary, the underestimation of occurrence by visual reports is probably due to their discontinuous nature and the difficulties in discriminate, at the ground, pure precipitation and blowing snow. Even though they underestimate occurrence, visual observations had a very low false alarm rate on occurrence.

It was shown that the MRR, whose lowest measurements are about 300 m above ground ($3^{rd}$ gate with a 100 m resolution),

is not able to measure blowing snow. This means that a configuration with a higher range resolution, at the expense of a lower maximum sampled height, must be used if this instrument is required to monitor blowing snow.

The measurements collected at DDU and illustrated in this paper show the potential of ground-based instruments to complement and validate satellite and numerical weather prediction model products related to precipitation. Such measurements can also provide information about the microphysical aspects of precipitation, like the dominant hydrometeor types and their

degree of riming in the present case. The synergy between remote-sensing and in-situ instruments has the potential to improve the quantification of snowfall amounts in conditions where blowing snow affects ground based measurements, even though much remains to be done in mixed cases with precipitation and blowing snow at the same time. The installation and long-term





operation of a similar combination of instruments should be conducted again, at DDU and at other locations in Antarctica.
Future work should focus on developing a better long-term constraints for radar-based snowfall retrievals because the one used
in this study was built on summer data only, on improving the quantitative discrimination between snowfall and blowing snow,
on the validation of satellite-based snowfall retrievals since it is of great interest to monitor the entire Antarctic continent, and
in further validating ERA-Interim reanalyses and other weather and climate models.

## 6   Data availability

All the relevant observations collected in the framework of the APRES3 project will be made available as soon as possible on
the website of the project (http://apres3.osug.fr).

*Acknowledgements.*  The authors are thankful to Météo France, and in particular to the team of Dumont d'Urville who provided the access
to their in-situ measurements and observations. We thank the French Polar Institute (IPEV), in particular Gregory Tran, Doris Thuillier, and
Patrice Godon, who allowed the APRES3 measurement campaign to take place. We thanks Paul Dufay, winteroverer at DDU, who provided
crucial assistance for the operation of the MRR during the winter season. The first author, Jacopo Grazioli, thanks the Swiss National
Science foundation SNF for the grant 513809, financing his participation to the project. The authors acknowledge also the support of the
French National Research Agency (ANR) to the APRES3 project. For the remote technical support provided, we want to thank Andrew
5  Pazmany and Johnatan Leachman (Prosensing Inc., manufacturer of MXPol). Jean-Baptiste Madeleine also thanks UPMC university for
financial assistance.



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
