# Peer review of "Measurements of precipitation in Dumont d'Urville, Terre Adélie, East Antarctica"

_The Cryosphere, 2017_

## Referee Comment (RC1) · Anonymous Referee #1 · 20 Mar 2017

**General comments:**

In the manuscript by Grazioli et al. the authors analyze a unique dataset of ground-based in-situ and remote sensing observations collected at Dumont d'Urville, Antarctica. The in-situ observations of an optical snow particle imager were used to derive statistics of the dominant particle types. A weighing gauge is used together with a vertically pointing rain radar to estimate snowfall rates and separating blowing snow from snowfall events. The one-year snowfall accumulation derived from the rain radar indicates a larger total accumulation as compared to ERA-Interim reanalysis data.

Although the total observation period is limited to only one year (some data only during summer period), I think this paper is an important contribution to characterize precipitation statistics and microphysical aspects of precipitation in Antarctica, a place where such observations are extremely rare. The analysis presented in this manuscript are certainly worth to be published in TC. Overall I find the paper well-structured but sometimes I think the English formulations and grammar could be further improved. I list some of them in the technical comments but I recommend an additional check by an English native speaker.

My major comment is related to the MRR relative calibration with the MXPol (section 2.2.1): You derive in Fig. 4 a linear correction for the MRR based on the assumption that the MXPol observations are correct. I assume that cm-sized snowflakes are rare in that area else you couldn't just compare X and K-Band reflectivities because the larger particles at K-Band would already deviate from Rayleigh scattering. I think you should mention this aspect somewhere in the text. Assuming the presence of mostly small particles, I am fine that no frequency correction is applied here. I am more curious to know how much you can be sure about the correct calibration of the MXPol? Did you perform any calibration with an external target (corner, sphere) for the MXPol? I think the question of the reliability of your Z-reference (MXPol) is quite important because an offset might cause a considerable bias in your MRR snowfall estimates whose deviation from ERA-Interim is one major conclusion of this paper. I think you should also discuss that more in the light of the findings in Palerme et al., TC, 2014 who found tha CloudSat snowfall estimates and occurrences and ERA-Interim agree quite well. I am aware that radar calibration is a delicate issue especially in such an environment but then you have to add this to your uncertainty estimate of your snowfall rates. One possible way of checking the absolute calibration is to use CloudSat overpass statistics since CloudSat is probably our currently best calibrated radar. Of course, the frequency difference and footprint is an issue, but one can use it as a sanity check to rule out larger biases. Examples how to do such a comparison are described for the MRR in Maahn et al., JGR, 2014 (in your reference list) or the earlier work for ground-based ARM radars by Protat et al., JTECH, 2010, 2011. I recommend to try such a calibration check if enough overpasses are available since it will enhance the confidence in your findings.

**Specific Comments:**
(Page, Line: Comment)

1, 4 and also 2, 32: Not clear what you mean with "model-free" here. I assume you mean more direct observations compared to re-analysis data, but also a Z-S relation is a certain kind of "model".

1, 7: "riming often occurs": "Often" is quite subjective and could for example also mean it occurs 90% of the time. Be more specific.

5, 44: "weather radar" is maybe a bit misleading. Cloud and precipitation radars are both observing "weather". I think the terms precipitation and cloud radar are better because they imply directly to

which particle sizes the radars are sensitive to. There are several studies using cloud radars which showed that cloud radars are sensitive to certain amounts of super-cooled liquid water (e.g. Shupe et al., JTECH, 2004).

Table 2: Just for clarification, I suggest adding a comment that all Z-S relations are derived for X-Band.

8, Table 2 and Z-S discussion: Did you consider to use your hydrometeor classification to derive separate Z-S relations for different hydrometeor mixtures. In that way you could present a solution to constrain the uncertainty range of future deployments by a combination of in-situ (or polarimetry) and vertically pointing radars.

9, 6: I am sceptical that you can say that you illustrate the "performance" of ERA-Interim when you compare it only at one grid box with pencil-beam observations. The model based method could perform over 99% of Antarctica well and just have an issue in this particular reagion which would mean that the model-based method is still quite good.

10, 14: "the most immediate microphysical parameter": I don't think hydrometeor type is more immediate than for example particle size or mass. Hydrometeor type is also quite ambiguously defined because what exactly is the definition of an aggregate or a rimed particle?

Figure 11b: I am wondering why the curve for the MRR is steadily increasing towards lower thresholds. Is that because the MRR noise is misinterpreted as precipitation signal below the MRR sensitivity limit?

17, 5-6: I find the argumentation a bit confusing. If it is known that blowing snow cannot reach more than 200m above ground, how should the MRR be able to see it with the first usable range gate at 300m? If you would configure it in a different way it probably would detect it. Apart from using a smaller range gate spacing you could also tilt the MRR to operate slant.

20, 49: I question that the Z-S relation would become less uncertain if you just use longer time series of in-situ data. In fact, the problem lies – as you mention – in the unknown variability of PSD and particle properties. I think it is worth mentioning that the addition of polarimetry or multi-frequency techniques are probably the currently only promising way to further constrain snowfall rate estimates.

**Technical Corrections/Typos:**

1, 4. Check whether "including of" is correct, I think "of" should be removed.

2, 38: I suggest removing "before anything else". Either there is a need or not.

3, 24: Rephrase "instruments were as illustrated in"

3, 25 and other occasions: I think you can remove "hereafter" after the instrument acronym.

5, 46: "for the day of the 15th December 2015": Why not simply "for the 15th December 2015"?

7, 62: "MRR radar": Leave "radar" it's already included in the acronym.

7, 68: and throughout the document: Sometimes you write "K-Band" with a dash, sometimes "X Band" without a dash. Sometimes you make the band letter italic sometimes not. Please make consistent throughout the text.

7, 74: Standard deviation of logarithmic reflectivities should be dB.

8, 96: "illustrates how the" -> "illustrates that the"

9, 3: Rephrase "as it is thought to provide" into for example "as it is considered to provide"

10, 19: Rephrase "instrument being close to the ground level, both…"

10, 23: Full stop after "height levels" and split in two sentences.

Figure 7: I suggest to add to the legend that height is above ground level and not above mean sea level

11, 30: "has been summarized" -> "is summarized"

11, 31: "hydrometeors too small" -> "hydrometeors being too small"

Figure 8: Explain the abbreviations used in the Pie-chart in the caption.

Figure 9: Legend indicates blue line color for Pluvio but in the plot it's black

13, 9: Rephrase "longer source of information"

13, 14: Remove "it" before "appears"

13, 17: Rephrase "this allows us to observe how the year under investigation had an extremely dry January"

15, 25 and other occasions: "cumulated" isn't accumulated more commonly used? Please check.

15, 25: The sentence is a bit complicated. To me it appears that you simply want to say that there are compensating errors that lead to the final agreement.

15, 34: I don't understand the "but" in this context.

16, 37: Remove "given the goal of the paper".

16, 38: "several times" please provide a range like "are conducted one to three times a day"

17, 11: Add a comma after "however"

18, 37: "underestimates"; split in two sentences after (illustrated in Fig. 11)

18, 4-5: Add a reference to Fig. 9 because it was not clear to me in which figure I am supposed to see the ERA underestimation after the discussion was mainly about overestimations by ERA.

---

## Referee Comment (RC2) · Anonymous Referee #2 · 16 Apr 2017

**General comments**

The paper by Grazioli et al "Measurements of precipitation in Dumont d'Urville, Terre Adélie, East Antarctica" presents analysis of new and unique in that region observations of precipitation at DDU station located in the coastal region of East Antarctica. The measurements include both in-situ (Pluvio, weather station) and remote sensing measurements (MRR and MxPol radars, lidar), which are compared to ERA-Interim reanalysis data. Such comprehensive measurements approach in order to quantify precipitation and also separate it from blowing snow on a long-term basis is much needed in Antarctica. This work contributes to an important question of quantification of precipitation over the Antarctic ice sheet and its representation by ERA-Interim reanalysis, which is commonly used for both analysis and as model forcing.

However, several important issues must be addressed before publication. The methodology has to be described in more details. The authors should not overstate their conclusions by saying that they have constrained estimates of precipitation in the region by using Z-S relationship derived from MRR and Pluvio during summer. A detailed description of calculating this local relationship is required in order to assess the validity of the results. Summer snowfall microphysical properties influencing radar Z can differ drastically from the rest of the year. Thus, using summer Z-S relationship can simply introduce a bias or a preference towards particle shapes and sizes observed in summer.

Methodology of data post-processing for each instrument and also their synergistic use has to be more detailed. Please see my specific comments.

**Specific comments:**

Abstract:
line 10 : "Climatological data..." - in what sense the term "climatological" is used? Are ERA-Interim data compared to observations for the year of measurements? Then "climatological" is inappropriate

Methods:
p. 3, line 18: "to name a few" - please name all relevant variables and their details

p. 3, line 26: MRR is not sensitive to clouds and sees only precipitation (it can be also precipitation not reaching the surface, i.e. "virga"). The authors should emphasize that MXPol is the only "weather radar" sensitive to both clouds and precipitation, while MRR should be rather called "precipitation radar". MRR will not only miss supercooled liquid clouds but also the ice clouds - except for those that are precipitating and thus can be classified as virga rather than clouds. These aspects have to be discussed when presenting the radars products.

p. 4, Fig. 1: add DDU location to the map

p. 4, line 35: What is the first useful range of MXPol radar? How is it influenced by the ground clutter?

p. 5, line 56: MASC camera has to be described in more details - what is the size range of detected particles? Are there any bias estimates due to the under-catch? What is the raw data processing methodology?

p. 5, line 59: details of the weather station? location, instruments, accuracy...

p. 7, line 81: "we optimized a local relation..." - this local Z-S relation is one of the most important milestones of this paper thus the authors have to explain in more details how it was calculated. The authors are saying that MRR and Pluvio measurements were used for calculating this Z-S relation. Why not using MASC which gives information about particle size and shape?

p. 8, Table 2: Please introduce all abbreviations from Matrosov 2009 reference.

The authors should justify why Matrosov 2009 relationships are applicable to DDU location. There are numerous other publications with various Z-S relationships.

p. 8, line 94-97: I would use MRR-detected precipitation as absolute truth with caution. MRR can easily miss very light ice precipitation to the ground, which in turn can be detected by Pluvio gauge. I recommend to do the following test: cluster Pluvio data by wind speeds with and without MRR-detected precip (precipitation at DDU is always accompanied by strong wind speeds thus is difficult to separate from blowing snow), and also check if "phantom" precip occurs just BEFORE MRR-detected precipitation.

p. 9: ERA-Interim data - is the grid closest to DDU location used or a mean of surrounding grids? Please give details

p. 10, line 15: the method of particle classification using MASC should be described in more details

p. 10, line 20: similarly, the classification method using MXPol should be described

p. 10, line 19: "The instrument being close to the ground level..." - at what height exactly agl MASC is located? This is important in interpretation of precip/blowing/drifting snow contribution

p. 11, fig 7: x-axis should be "fraction of occurrence, %"

p. 11: I don't agree with the statement that the majority of small particles can be assumed to be blowing snow particles picked up from the surface. Precipitating particles of small sizes are also common in Antarctica. And I expected the

observations described here to shed more light into this issue rather than using common assumptions.

Contribution of blowing snow to the small particles can be estimated by separating MASC-analyzed particles for clear-sky blowing snow events and precipitation events.

p 12, Fig 8: How is the riming index obtained? Please add units and values to the y-axis. All abbreviations used in naming snow particles in the pie-chart must be explained. "The riming index is undefined for "small" particles" - what is the threshold for defining "small" particles?

p. 12, line 53: "...though the curves do not co-fluctuate well". Please specify what do you mean. What is the correlation coefficient?

p. 12, line 54: "this optimized Z-S relation provides estimate that are close to the B90A relation.." Specify what is the B90A relation of Matrosov 2009. Why is the local Z-S relation close particularly to B90A? In Table 2, the reference is Matrosov et al 2009, while here Matrosov 2009. Please correct.

To my opinion, the authors cannot say that they have reduced the uncertainty in quantifying year-long snowfall rates by applying local optimized relationship calculated using only summer measurements. In order to justify this, seasonal evolution of cloud particles has to be analyzed. While this is a subject of future analysis, the authors should discuss limitations of their method.

p. 13, fig 9: Caption should be completed and self-explanatory

p. 16, line 45 and section 4.2: MRR cannot capture blowing snow... I am asking the authors to clarify this issue and show more clearly using other measurements (Pluvio, AWS) that MRR sees nothing during clear-sky blowing snow events

**Technical corrections:**

p. 2, line 31: e.g., Konig-Langlo et al.. (as this is not the only paper about human meteorological measurements)

p. 2, line 37: "for the medium..." - delete 'for'

p. 12 and throughout the text: "cumulated" -> either accumulated or cumulative

p. 12, lines 4-5: "within what could be observed...." - the whole sentence should be rephrased

p. 14, Table3: units? mm we?

---

## Author Comment (AC1) · 19 May 2017

**tc-2017-18**

J. Grazioli, et al

May 18, 2017

**Responses**

With the present supplementary document we provide our responses to the comments of the two anonymous reviewers of the manuscript tc-2017-18, entitled "Measurements of precipitation in Dumont d'Urville, Terre Adélie, East Antarctica".

The comments of the reviewers are reported in *italic* font. Quotations of the manuscript in its revised or in its original form are reported in blue. The figures used to support our answers to the reviewers can be found at the end of this document.

**Anonymous reviewer 1**

We would like to start by thanking the reviewer for their valuable contribution and constructive comments that lead, in our opinion, to an improvement of the quality of the manuscript.

**General comments**

In the manuscript by Grazioli et al. the authors analyze a unique dataset of ground-based in-situ and remote sensing observations collected at Dumont dUrville, Antarctica. The in-situ observations of an optical snow particle imager were used to derive statistics of the dominant particle types. A weighing gauge is used together with a vertically pointing rain radar to estimate snowfall rates and separating blowing snow from snowfall events. The one-year snowfall accumulation derived from the rain radar indicates a larger total accumulation as compared to ERA-Interim reanalysis data. Although the total observation period is limited to only one year (some data only during summer period), I think this paper is an important contribution to characterize precipitation statistics and microphysical aspects of precipitation in Antarctica, a place where such observations are extremely rare. The analysis presented in this manuscript are certainly worth to be published in TC.

We thank the reviewer for this very positive comment. We are confident that our experience is indeed a valuable example for other installations in Antarctica.

Overall I find the paper well- structured but sometimes I think the English formulations and grammar could be further improved. I list some of them in the technical comments but I recommend an additional check by an English native speaker.

Before submission we conduct an internal review with a native speaker colleague. We will carefully check the quality of the revised version of the manuscript as well, that will eventually be edited by the Copernicus specialized staff.

My major comment is related to the MRR relative calibration with the MXPol (section 2.2.1): You derive in Fig. 4 a linear correction for the MRR based on the assumption that the MXPol observations are correct. I assume that cm-sized snowflakes are rare in that area else you couldn't just compare X and K- Band reflectivities because the larger particles at K-Band would already deviate from Rayleigh scattering.

As the reviewer suggests, our method of correction intrinsically suggests that cm-size snowflakes leading to non-Reyleigh effects are rare. We would like to remind that our approach is to base everything on the observations. If cm-size snowflakes would have been frequent, probably the relation in Fig.4 would not be linear (or in other words, the linear relation would have not be such a good fit). Additionally, the linear relation that we obtained is practically a simple additive offset (slope very close to one) of 6.14 dB, which leads us to assume that radome attenuation was causing the differences in reflectivity among the two instruments.

We recently conducted a second field campaign in DDU, and for a test period in Jan/Feb 2017, a second MRR was deployed outside the radome. The comparison among the two instruments (Fig. 1 at the end of this document) showed a similar offset (6.94 dB). These observations: (i) confirm the effect of the radome in term of attenuation, (ii) make us confident about the magnitude of our correction, (iii) give us also an indication about the goodness of calibration of MXPol. Please also note that the difference between the two offsets (the one calculated with MXPol and the one calculated with the second MRR) is within the uncertainty of our conversion relation, as given in Eq. 1 of the manuscript.

I think you should mention this aspect somewhere in the text. Assuming the presence of mostly small particles, I am fine that no frequency correction is applied here.

**We rephrased a sentence in the text as:**

Because overall the relation between the two sets of measurements is close to linear ( $\rho^2 \approx 0.88$ ), and almost equivalent to a simple offset subtraction, we can hypothesize that eventual non-Rayleigh effect, due to cm-size snowflakes, were similar at the two frequencies the following conversion has been applied to MRR data...

I am more curious to know how much you can be sure about the correct calibration of the MXPol? Did you perform any calibration with an external target (corner, sphere) for the MXPol? I think the question of the reliability of your Z-reference (MXPol) is quite important because an offset might cause a considerable bias in your MRR snowfall estimates whose deviation from ERA-Interim is one major conclusion of this paper.

About the calibration of MXPol:

- MXPol has been technically inspected before the measurement campaign. In that occasion we conducted mostly measurements of the transmitting path. The absolute calibration of the radar, as provided by the manufacturer, has been refined (≈3 dB correction) through a long-term comparison with several ground observations of drop size distributions by means of Parsivel disdrometers. This adapted calibration constant has been validated after the measurement campaign by measuring the reflectivity of a balloon-lifted metal sphere of known characteristics. However, as the radar has been dismantled for the transport from/to Antarctica, we cannot guarantee that the calibration values obtained in Europe were perfectly valid in Antarctica. However (next bullet)
- Calibration has here a lower weight than what mentioned by the reviewer. In fact, because we were aware of these uncertainties, the final Z S relation has been obtained by local comparison with the Pluvio2 data. We think it is important to try the best to have correct Z values (reason why we use MXPol to correct for the radome attenuation on MRR measurements), but practically the retrieved S values are locally trained on the available Z.
- As mentioned just above, following what has been shown in Fig. 1 at the end of this document, the new available 2017 data (from the second MRR) tend to confirm the magnitude of the correction and thus, indirectly, the calibration of MXPol.

I think you should also discuss that more in the light of the findings in Palerme et al., TC, 2014 who found that CloudSat snowfall estimates and occurrences and ERA-Interim agree quite well. I am aware that radar calibration is a delicate issue especially in such an environment but then you have to add this to your uncertainty estimate of your snowfall rates. One possible way of checking the absolute calibration is to use CloudSat overpass statistics since CloudSat is probably our currently best calibrated radar. Of course, the frequency difference and footprint is an issue, but one can use it as a sanity check to rule out larger biases. Examples how to do such a comparison are described for the MRR in Maahn et al., JGR, 2014 (in your reference list) or the earlier work for ground-based ARM radars by Protat et al., JTECH, 2010, 2011. I recommend to try such a calibration check if enough overpasses are available since it will enhance the confidence in your findings.

Satellite observations are indeed very important for the APRES3 project (as the name of the project says: Antarctic Precipitation, Remote Sensing from Surface and **Space**), and ongoing research is focussing on the subject. We believe that what the reviewer suggests is outside the goal of the present manuscript, not only for the complexity of the comparison (different sampling volumes, blind zone, sampling effect, to cite some issues) but also because larger datasets (more years, more seasons, as in Maahn et al. (2014); Palerme et al. (2014)) are, in our opinion, needed to derive robust statistics.

Most importantly, probably we already addressed the concerns about possible large biases in our measurements, thanks to the observations conducted with the second MRR in 2017 (described just above). For the calibration of the MRR we are in fact more confident in the measurements of a co-located similar instrument covering the same sampling volumes.

**Specific comments**

1. 1, 4 and also 2, 32: Not clear what you mean with model-free here. I assume you mean more direct observations compared to re-analysis data, but also a Z-S relation is a certain kind of "model".

In this case we meant "based on observations". We rephrased 1,4 as:

These instruments collected the first measurements of precipitation in the region of Terre Adélie (Adélie Land) not based on numerical weather models, including precipitation microphysics. And 2, 32 as: Recent research proposed a climatology of precipitation over a large part of the continent (Palerme et al., 2014, 2016) by exploiting the potential of the profiling radar on-board the CloudSat satellite

2. 1, 7: "riming often occurs": "Often" is quite subjective and could for example also mean it occurs 90% of the time. Be more specific.

We rephrased this part of the sentence as:

...riming is a recurring process. 11% of the measured particles were fully developed graupel, and aggregates had a mean riming degree of about 30%.

3. 5, 44: "weather radar" is maybe a bit misleading. Cloud and precipitation radars are both observing weather. I think the terms precipitation and cloud radar are better because they imply directly to which particle sizes the radars are sensitive to. There are several studies using cloud radars which showed that cloud radars are sensitive to certain amounts of super-cooled liquid water (e.g. Shupe et al., JTECH, 2004).

We used now the term "radar" or "polarimetric radar". The work mentioned by the reviewer was in fact using much higher (94 GHz) frequency than the radars used in our research work.

4. Table 2: Just for clarification, I suggest adding a comment that all Z-S relations are derived for X-Band.

The information has been added in the caption.

5. 8, Table 2 and Z-S discussion: Did you consider to use your hydrometeor classification to derive separate Z-S relations for different hydrometeor mixtures. In that way you could present a solution to constrain the uncertainty range of future deployments by a combination of in-situ (or polarimetry) and vertically pointing radars.

The reviewer is right. The synergy of the instruments suggests that this approach could be tried. In our case, because we had only few months of co-location of MASC and MRR (i.e., we would not be able to select among different Z-S relations for the periods when MASC and MXPol were not deployed), we decided to calibrate an "average" relation. To be fair, as stressed also by reviewer 2, even in this case the goodness of the relation remains uncertain for the winter months, where the MRR was the only instruments deployed.

The suggestion of the reviewer remains however very good. The development of a hydrometeor-dependent relation should be tried as soon as long series (covering different seasons) of co-located measurements will be available.

6. 9, 6: I am sceptical that you can say that you illustrate the performance of ERA-Interim when you compare it only at one grid box with pencil-beam observations. The model based method could perform over 99% of Antarctica well and just have an issue in this particular region which would mean that the model-based method is still quite good.

We agree with the concerns of the reviewer, and we rephrased the sentence avoiding to refer to "performance":

ERA-Interim reanalysis is used here for this reason, and because of its global coverage and easy access.

Throughout the manuscript, we rephrased the parts that could lead to similar concerns. Our objective is not to validate ERA Interim over the continent, but to show how the available reanalysis compare to our first measurements of precipitation collected in this specific area.

7. 10, 14: "the most immediate microphysical parameter": I dont think hydrometeor type is more immediate than for example particle size or mass. Hydrometeor type is also quite ambiguously defined because what exactly is the definition of an aggregate or a rimed particle?

We agree that our statement is too subjective. We propose the following rephrasing:

it is worth to investigate an important microphysical parameter:

8. Figure 11b: I am wondering why the curve for the MRR is steadily increasing towards lower thresholds. Is that because the MRR noise is misinterpreted as precipitation signal below the MRR sensitivity limit?

This is a good question, and we investigated this aspect more closely. The MRR noise is actually satisfactorily censored by the method of Maahn and Kollias (2012), used to process the raw MRR data. The behaviour observed by the reviewer results from the fact that, in the curves of Figure 11, the snowfall intensity is calculated at the 6 h scale, thus increasing the sensitivity of the MRR. For example, let us assume that a snowfall events of 30 min duration, with an intensity of 0.1 mm h-1 (roughly 5 dBZ at X band for aggregates) occurs. At 6 h time scale this results in an intensity of about 0.008 mm h-1.

We agree with the reviewer that the thresholds we show in the figure are too low to be realistic or reasonable (down to  $10^{-5}$ ), and thus in the revised version we increase the minimum one to  $10^{-3}$  mm h-1.

9. 17, 5-6: I find the argumentation a bit confusing. If it is known that blowing snow cannot reach more than 200m above ground, how should the MRR be able to see it with the first usable range gate at 300m? If you would configure it in a different way it probably would detect it. Apart from using a smaller range gate spacing you could also tilt the MRR to operate slant.

We agree with the reviewer and we simplified the sentence as:

confirms that the current MRR configuration does not capture the occurrence of blowing snow events. Lower range gate spacing should be employed (lower than 100 m used in the measurements shown here) if blowing snow is of interest.

Because we have now a continuous (and unprecedented) monitoring period with this MRR, we are reluctant to change its configuration. However, since February 2017, a second MRR running at a resolution of 15 m has been also installed in DDU. When enough data of both MRRs will be collected, we will be able to better focus on blowing snow in the next steps of our research.

10. 20, 49: I question that the Z-S relation would become less uncertain if you just use longer time series of in-situ data. In fact, the problem lies as you mention in the unknown variability of PSD and particle properties. I think it is worth mentioning that the addition of polarimetry or multi-frequency techniques are probably the currently only promising way to further constrain snowfall rate estimates.

We rephrased this sentence, to stress the importance to better make use of the synergy of microphysical observations:

Future work should focus on developing a better long-term constraints for radar-based snowfall estimations by means of in-situ measurements of precipitation in synergy with microphysical observations and retrievals, because the relation used in this study was built on summer data only, on better discriminating between snowfall.

We agree with the reviewer about the potential of polarimetry and dual-frequency. We are also aware, however, that logistics in Antarctica is extremely complex and demanding, and it is hard to imagine (at the present days) such complex radar systems run unsupervised for several months, as the MRR did. These techniques are only at the stage of research in more comfortable regions. And this is even more difficult in Antarctica. However, this type of campaign, if it is repeated several times, would enable us to construct little by little the long time series for which the reviewer refers.

**Technical Corrections/Typos**

- 1. 1, 4. Check whether "including of" is correct, I think "of" should be removed. Rephrased as: ...including precipitation microphysics.
- 2. 2, 38: I suggest removing "before anything else". Either there is a need or not.
   We agree with the reviewer. Rephrased as: There is therefore the need for accurate measurements of precipitation...
- 3. 3, 24: Rephrase "instruments were as illustrated in"

The sentence has been rephrased as: The instruments were deployed as illustrated...

- 4. 3, 25 and other occasions: I think you can remove "hereafter" after the instrument acronym. We agree. The acronyms are now defined in parentheses the first time they appear, without adding "hereafter".
- 5. 5, 46: "for the day of the 15th December 2015": Why not simply "for the 15th December 2015"? Indeed. Rephrased, according to the suggestion.
- 7, 62: "MRR radar": Leave "radar" its already included in the acronym. We agree. Changed (also the title of the section), accordingly.
- 7. 7, 68: and throughout the document: Sometimes you write "K-Band" with a dash, sometimes "X Band" without a dash. Sometimes you make the band letter italic sometimes not. Please make consistent throughout the text.

We thank the reviewer, who spotted this typo. The convention has been harmonized throughout the manuscript, as follows: "X-band", and "K-band".

- 7, 74: Standard deviation of logarithmic reflectivities should be dB. Changed, from dBZ to dB, according to the suggestion.
- 9. 8, 96: illustrates how the → "illustrates that the" Rephrased, according to the suggestion of the reviewer.
- 10. 9, 3: Rephrase "as it is thought to provide" into for example "as it is considered to provide" Rephrased, according to the suggestion of the reviewer.
- 11. 10, 19: Rephrase "instrument being close to the ground level, both..." Rephrased as: Because the instruments are close to the ground level, both precipitation
- 12. 10, 23: Full stop after "height levels" and split in two sentences. The sentence has been split in two, as suggested.
- 13. Figure 7: I suggest to add to the legend that height is above ground level and not above mean sea level The legend now reads: ... height above ground ...
- 14. 11, 30: "has been summarized"  $\rightarrow$  "is summarized" Changed, according to the suggestion.
- 15. 11, 31: "hydrometeors too small"  $\rightarrow$  "hydrometeors being too small" Changed, as suggested.
- 16. Figure 8: Explain the abbreviations used in the Pie-chart in the caption. We added the following sentence in the Pie-chart:

The classes of the chart are: small particles (SP), columnar crystals (CC), aggregates (AG), planar crystals (PC), graupel (GR), combination of columnar and planar crystals (CPC), as described in Praz et al. (2017).

17. Figure 9: Legend indicates blue line color for Pluvio but in the plot its black

We think that the confusion originates by the fact that the blue Pluvio2 line was shown only in the bottom panel (summer campaign 2015/2016) and it was very close to the red line of the optimized Z - S relation. We built now a complete legend in the top panel, and we thickened the blue line for a better visualization.

 13, 9: Rephrase "longer source of information" We removed "longer" from this sentence.

- 19. 13, 14: Remove "it" before "appears" Done.
- 20. 13, 17: Rephrase "this allows us to observe how the year under investigation had an extremely dry January"

We simplified the sentence as:

The year under investigation had an extremely dry January, and an extremely snowy September...

21. 15, 25 and other occasions: "cumulated" isnt accumulated more commonly used? Please check.

Here the reviewer probably refers to our choice of words when we say "cumulated precipitation" instead than "accumulated precipitation". This was done on purpose, to underline the fact that we measure precipitation and cumulate its values. With the term "accumulation" we may generate misunderstandings for the scientific community devoted to ice mass balance as it is usually interpreted as ground accumulation (resulting from precipitation, wind transport, vapour deposition, and sublimation).

22. 15, 25: The sentence is a bit complicated. To me it appears that you simply want to say that there are compensating errors that lead to the final agreement.

We wanted to express a slightly different concept: the overestimation of occurrence (of lower intensity events) compensates the underestimation of the contribution of the most intense events. We rephrased as:

The overestimation of occurrence compensates the underestimation of the most intense snowfall events, such that at the end of January the total cumulated precipitation of ERA-interim gets close to the one of the Pluvio2.

23. 15, 34: I dont understand the "but" in this context.

We substituted it with "and". Our intention was to say that not only visual observations of precipitation are archived, but also visual observations of clouds and blowing snow.

24. 16, 37: Remove "given the goal of the paper".

Done.

25. 16, 38: "several times" please provide a range like "are conducted one to three times a day" We rephrased the sentence taking into account this remark:

The observations are conducted during each day, on average every 5h (with higher frequency during day hours), and we compare

26. 17, 11: Add a comma after "however"

The word 'however" has been removed from the revised sentence.

- 27. 18, 37: "underestimates"; split in two sentences after (illustrated in Fig. 11) Revised, according to the suggestion.
- 28. 18, 4-5: Add a reference to Fig. 9 because it was not clear to me in which figure I am supposed to see the ERA underestimation after the discussion was mainly about overestimations by ERA.
  We added the reference, as suggested.

**Anonymous reviewer 2**

Thanks to several insightful comments of the reviewer, we could fix some weaknesses in the manuscript and provide a more objective discussion of the results.

**General comments**

The paper by Grazioli et al "Measurements of precipitation in Dumont d'Urville, Terre Adlie, East Antarctica" presents analysis of new and unique in that region observations of precipitation at DDU station located in the coastal region of East Antarctica. The measurements include both in-situ (Pluvio, weather station) and remote sensing measurements (MRR and MxPol radars, lidar), which are compared to ERA-Interim reanalysis data. Such comprehensive measurements approach in order to quantify precipitation and also separate it from blowing snow on a long- term basis is much needed in Antarctica. This work contributes to an important question of quantification of precipitation over the Antarctic ice sheet and its representation by ERA-Interim reanalysis, which is commonly used for both analysis and as model forcing.

We are very thankful to the reviewer, who recognized the originality of our work.

However, several important issues must be addressed before publication. The methodology has to be described in more details. The authors should not overstate their conclusions by saying that they have constrained estimates of precipitation in the region by using Z-S relationship derived from MRR and Pluvio during summer. A detailed description of calculating this local relationship is required in order to assess the validity of the results. Summer snowfall microphysical properties influencing radar Z can differ drastically from the rest of the year. Thus, using summer Z-S relationship can simply introduce a bias or a preference towards particle shapes and sizes observed in summer.

We understand, and partially agree with the concerns of the reviewer. We believe we addressed his/her main points in the general comments below.

Methodology of data post-processing for each instrument and also their synergistic use has to be more detailed. Please see my specific comments.

We provided additional information for some methodologies, especially: (i) the MASC classification, (ii) MXPol classification and, (iii) the calibration of a local Z-S relation (see answer to the specific comments below). We especially clarified that (i) and (ii) are published and open access works (Grazioli et al., 2015; Praz et al., 2017), and therefore the technical details are easily accessible, while we included explicitly the details of (iii).

**Specific comments**

1. Abstract: line 10 : "Climatological data..." - in what sense the term "climatological" is used? Are ERA-Interim data compared to observations for the year of measurements? Then "climatological" is inappropriate

We rephrased this sentence as follows:

Data obtained in previous research from satellite-borne radars, and the ERA-Interim reanalysis of the European Center for Medium -Range Weather Forecasts (ECMWF) both provide...

In this way we clarify that the climatological data we refer in this specific point are not the ground-based observations of this paper, but the ones obtained using CloudSat and presented in Palerme et al. (2014).

- 2. Methods: p. 3, line 18: "to name a few" please name all relevant variables and their details Following the suggestion of the reviewer, we named the following variables: (temperature, wind speed, wind direction, relative and specific humidity, atmospheric pressure)
- 3. p. 3, line 26: MRR is not sensitive to clouds and sees only precipitation (it can be also precipitation not reaching the surface, i.e. "virga"). The authors should emphasize that MXPol is the only "weather radar" sensitive to both clouds and precipitation, while MRR should be rather called "precipitation radar". MRR

will not only miss supercooled liquid clouds but also the ice clouds - except for those that are precipitating and thus can be classified as virga rather than clouds. These aspects have to be discussed when presenting the radars products.

Because the term "clouds" was generating probably confusion in this sentence, we rephrased it as:

This radar system is used to vertically profile precipitation with a resolution of  $100 \,\mathrm{m}$

However, we respectfully disagree about this distinction. Both radars, given their frequency and sensitivity, are first of all measuring precipitation-size hydrometeors, and can only occasionally measure ice-phase clouds. In our view, the clear distinction proposed by the reviewer is not applicable in this case.

4. p. 4, Fig. 1: add DDU location to the map

DDU was already located with a green filled circle (as explained in the caption). We added the acronym "DDU" next to the filled circle.

5. p. 4, line 35: What is the first useful range of MXPol radar? How is it influenced by the ground clutter?

As MXPol is a scanning radar and not a profiler, the influence of clutter depends on the direction of transmission, and therefore is not directly related to the first useful range. The blind range of MXPol is about 250 m, as the MRR. In the present work, MXPol is used to produce Fig. 4 (comparison with MRR), Fig. 6 (classification example), and Fig. 7 (hydrometeor type statistics during Summer 2015/2016).

- Fig. 4: Only MXPol measurements that overlap with MRR measurements are taken, thus at distances larger than the blind range of MXPol. The caption has been clarified as: Scatter plot of reflectivity values at 9.41 GHz (X-band, measured by MXPol) and at 24.3 GHz (K-band, measured by the MRR) during the summer campaign 2015/2016. The data correspond to time-steps when both radar were profiling (PPI and RHI scans of MXPol do not contribute).
- Fig. 6: Here it is visible some residual noise in the classification due to the clutter, mostly affecting the elevation angles lower than 2°. An explanation has been added in the caption, that indeed needed further clarification:

Noise in the classification at the lowest elevation angles is due to ground clutter. Range gates closer than 2 km with respect to the radar location have been censored to allow reliable polarimetric variables to be computed. Elevation angles larger than  $45^{\circ}$  have been censored as well in order to limit the geometric reduction of the intensity of polarimetric signature with increasing elevation angles (Ryzhkov et al., 2005).

- Fig. 7 The statistics are calculated with the same constraints clarified in Fig. 6. Additionally, elevation angles lower than 3° are censored, and only heights above 400 m are shown in this figure.
- 6. p. 5, line 56: MASC camera has to be described in more details what is the size range of detected particles? Are there any bias estimates due to the under-catch? What is the raw data processing methodology?

We rephrased this part of the paragraph, by providing additional description (resolution, measurement principle), as:

This instrument collects high-resolution stereoscopic photographs of snowflakes in free fall, while they cross its sampling area (Garrett et al., 2012), thus providing information about snowfall microphysics and particle fall velocity. The MASC was using three identical 2448 x 2048 pixels cameras (with common focal point) with apertures and exposure times adjusted to trade off between the contrast on snowflakes photographs and motion blur effects, and a resolution of about 33  $\mu$ m per pixel. The cameras are triggered when a falling particle crosses two series of near-infrared sensors. A detailed description of the system and its calibration can be found in Garrett et al. (2012); Praz et al. (2017).

We believe that for a more in-depth technical overview, an interested reader should refer to the open access articles that we cite in this paragraph.

7. p. 5, line 59: details of the weather station? location, instruments, accuracy...

The reviewer is right, we did not provide enough information for the reader to understand exactly which instrument was mentioned, and where it was exactly installed. We rephrased as:

To complete the set of in-situ measurements, a weather station (Vaisala Weather Transmitter WXT 520) was installed close to the Pluvio2 and the MASC, to sample the environmental conditions in the close proximity of their measurements, as illustrated in Fig. 2.

As the instrument and its manufacturer are relatively well known, we believe that additional information for example about its accuracy are not needed here. Also Table 1 was edited, adding the precise name of the instrument.

8. p. 7, line 81: "we optimized a local relation..." - this local Z-S relation is one of the most important milestones of this paper thus the authors have to explain in more details how it was calculated.

We agree with the reviewer. We added some details about the relation, and this part of the manuscript now reads:

For this reason, we optimized a local power law, by fitting its two parameters in the Z-S space given by the MRR measurements at the lowest available height and the Pluvio2measurements collected close to the ground, during the Summer period 2015/2016. The parameters (intercept and exponent) of the power law are obtained by means of nonlinear least square estimation. The local relation, also listed in Table 2, takes the form of  $Z = 75.85 S^{0.91}$ . In order to mitigate the difference in sampling volume of the two instruments, it has been derived for hourly data. The 95% confidence intervals for the two parameter are 68.68-82.94 and 0.776-1.09 respectively.

The authors are saying that MRR and Pluvio measurements were used for calculating this Z-S relation. Why not using MASC which gives information about particle size and shape?

Using the MASC is a different, and potentially valid approach. Ongoing research is indeed focussed on the added value of the MASC for scattering simulations, but this work is still preliminary. In our case, because we had only few months of co-location of MASC and MRR (i.e., we would not be able to select among different Z-S relations for the periods when MASC and MXPol were not deployed), we decided to calibrate an "average" relation using the Pluvio2. We added a sentence in the conclusions to open the perspective of a better synergistic use of MASC and MRR, when more co-located observations will be available:

Efforts will be devoted to develop a better long-term constraint for radar-based snowfall estimations by means of in-situ measurements of precipitation in synergy with microphysical observations and retrievals, because the relation used in this study was built on summer data only.Future work should also focus on better discriminating between snowfall...

Additionally, the  $Pluvio^2$  directly measures the equivalent liquid water amount, while the MASC provides information only about the geometry of the particles

9. p. 8, Table 2: Please introduce all abbreviations from Matrosov 2009 reference.

We believe that it may be enough to explain why there are different relations, and then refer to the original manuscripts for details. We added the following clarification to the caption:

The 6 X-band relations originate from two different datasets (B90, ground-based, and W08, from in-situ aircraft measurements), and three different mass to diameter relations, as detailed in Matrosov et al. (2009)

The authors should justify why Matrosov 2009 relationships are applicable to DDU location. There are numerous other publications with various Z-S relationships.

The fact that we are not sure if these relations are applicable (relations for Antarctica do not exist yet) led us indeed to build a local relation, and not to use them directly. We carefully checked the manuscript to make sure the reader is not brought to think that we imply that they are applicable explicitly for Antarctica. The relations of Matrosov were chosen because they covered different types of snow (simulations obtained from ground-based and in-situ microphysical observations), at X-band (MXPol frequency). To our knowledge, these are among the few Z-S relations available in the literature that are explicitly developed for X-band, and the most recent ones (see Scipion et al., 2013, for an overview).

10. p. 8, line 94-97: I would use MRR-detected precipitation as absolute truth with caution. MRR can easily miss very light ice precipitation to the ground, which in turn can be detected by Pluvio gauge.

We thank the reviewer for this input. We carefully checked, once more, the available literature and we could not find research supporting this statement. As an order of magnitude, with the MRR processing method of Maahn and Kollias (2012) snowfall rates on the order of around  $0.05 \,\mathrm{mm}\,\mathrm{h}^{-1}$  (0.01 in the paper, but we add here the radome attenuation) can be measured, at the temporal scale of one minute. By taking the most optimistic specifications of the manufacturer of the Pluvio2 (i.e.  $0.01 \,\mathrm{mm}\,\mathrm{min}^{-1}$ ), we obtain minimum detectable intensity of  $0.6 \,\mathrm{mm}\,\mathrm{h}^{-1}$ , over the same time scale. It is of course an over simplification, but it excludes the fact that a-priori Pluvio2 is more sensitive than the MRR.

I recommend to do the following test: cluster Pluvio data by wind speeds with and without MRR-detected precip (precipitation at DDU is always accompanied by strong wind speeds thus is difficult to separate from blowing snow), and also check if "phantom" precip occurs just BEFORE MRR- detected precipitation.

Before to answer to the specific point, we would like to underline a conceptual change that we implemented in the revised manuscript. We originally believed that the phantom accumulations observed were mostly associated with blowing snow, but they were in reality due to the vibrations of the instrument (still happening because of strong winds). In the revised manuscript we took care of defining the pretreatment of the Pluvio2 data without implying that we could measure clear-sky blowing snow. We came to this conclusion by observing how many images (almost none) were collected by the MASC during the censored time-steps. We rephrased the section describing the pre-treatment as:

In such cases, no precipitation was observed by the researchers that were present on site and no precipitation signal was visible in the MRR data but the content of the Pluvio2 bucket was increasing. In order to censor these cases, we combined the information coming from remote sensing (MRR) and in-situ data (Pluvio2). More precisely, time steps when no signal was recorded by the MRR at its lowest available gate (300 m above ground level), are considered "precipitation free" and any increase in the cumulative precipitation records of the Pluvio2 is thus related to external contaminations. The assumption is that precipitation is extremely unlikely to completely develop in the lowest 300 m of the atmosphere. An example of the behaviour of this simple censoring filter can be found in Fig. 5 (a). From the end of October 2015 until the end of January 2016, about 14 mm of liquid water equivalent snowfall have been removed, corresponding to about 21% of the uncensored data.

Figure 5 (b) shows the evolution of wind speed in the near proximity of the Pluvio2 inlet and illustrates that the most intense phantom accumulations occur when the strongest wind peaks are observed. Because the cameras of the co-located MASC were not triggered by hydrometeors during the censored time-steps, we ruled out the possibility that phantom accumulation is in this case due to clear-sky blowing snow, and we hypothesize that it is caused by wind-induced vibrations of the instrument. It must be noted that this simple pre-treatment cannot compensate the contribution of snowfall mixed with blowing snow, when the positive contribution of blowing snow and precipitation, and the negative contribution due to wind-induced loss of catching efficiency occur together.

In addition, we rephrased all our previous claims about potential blowing snow estimation in Sec. 4.2 and in the conclusions.

We move now to the suggestion of the reviewer. Figure 2 at the end of this document shows the effect of our filter on the scatterplot of wind (Vaisala station) and precipitation (Pluvio2). We can observe that our filter removes the wind-based dependency that appears in unfiltered data. In our opinion this is a positive behaviour.

About the second suggestion (i.e., check if phantom precip occurs just before MRR-detected precipitation): this was a good idea, and something we did not focus on. We visually inspected the MRR measurements for all the days that had "phantom" Pluvio2 accumulation, and a pattern as hypothesized by the reviewer could not be found. These accumulations occur before or after precipitation, and even during sunny days where precipitation was not observed at all in the MRR data (confirmed by the site notes collected on the field). As Fig. 5 in manuscript suggests (and explained just above), this phenomenon seems to be related to the wind.

We do not fully agree, however, with the statement that "precipitation in DDU always occur with strong winds". An example can be found in Fig. 5 of the manuscript where we can observe several precipitation events occurring with winds lower than  $10 \,\mathrm{ms}^{-1}$ .

11. p. 9: ERA-Interim data - is the grid closest to DDU location used or a mean of surrounding grids? Please give details

The closest grid point is used. In the case of DDU, the center of the closest grid point (66.75S, 140.25E) is also quite close to the actual location of the scientific base. We rephrased as follows:

The analyses at 00 UTC and 12 UTC, and forecast time steps of 6, and 12 h are used in the present work for the grid point which is the closest to DDU.

- 12. p. 10, line 15: the method of particle classification using MASC should be described in more details Please, refer to our answer to the specific comment number 6, above.
- 13. p. 10, line 20: similarly, the classification method using MXPol should be described

We expanded the description of the method, as follows:

A second classification method is obtained from the polarimetric data of MXPol, that can be converted into hydrometeor measurements with an hydrometeor classification algorithm (Grazioli et al., 2015). This algorithm was developed by partitioning a large number of radar observations into spatially coherent clusters by means of data mining techniques, and then to assign to each cluster a dominant hydrometeor type by means of scattering simulations, interpretation of polarimetric signatures, and comparison with in-situ data.

Of course, for a full comprehension of the method, the interested reader needs to refer to the open access article Grazioli et al. (2015).

14. p. 10, line 19: "The instrument being close to the ground level..." - at what height exactly agl MASC is located? This is important in interpretation of precip/blowing/drifting snow contribution

The MASC was installed about 2 to 3 m above ground. This is now explicitly mentioned in the manuscript.

- 15. p. 11, fig 7: x-axis should be "fraction of occurrence,"We corrected the figure, according to the suggestion of the reviewer.
- 16. p. 11: I don't agree with the statement that the majority of small particles can be assumed to be blowing snow particles picked up from the surface. Precipitating particles of small sizes are also common in Antarctica. And I expected the observations described here to shed more light into this issue rather than using common assumptions. Contribution of blowing snow to the small particles can be estimated by separating MASC-analyzed particles for clear-sky blowing snow events and precipitation events.

We observed a much larger percentage of small particles for the measurement period in Antarctica, compared to a long deployment of the instrument in a wind-sheltered location in the Swiss Alps, and we have seen that this contribution was associated to some events (strong wind, impressive blowing snow observed by the researchers on site, a lot of particles recorded by the MASC), during which small particles constituted almost 100% of the measurements. We rephrased this part of the manuscript as:

At ground level, the majority of the particles (54%) are classified as "small", indicating hydrometeors being too small for their geometry and texture to be properly captured by the MASC. This proportion is three times higher than similar measurements collected in a wind sheltered location in the Swiss Alps, while the proportion of the other hydrometeors is similar among the two different locations (not shown here). The occurrence of strong katabatic winds being a major difference between the sites, it can be assumed that the large majority of these "small particles" observed at DDU is associated with blowing snow. During blowing snow events with strong winds (identified from visual and MRR observations on site), the number of images collected by the MASC is very large. The majority of those being classified as "small particles", this results in a large percentage of this hydrometeor type in the final statistics.

We would like to show to the reviewer two contrasted (but not excessively) cases, that illustrate this effect. Fig. 3 and 4 at the end of this document show the evolution of wind speed, number of particles recorded by the MASC, and their classification for the 11.11.2015 (precipitation with strong wind, blowing snow observed consistently on-site), and the 15.12.2015 (precipitation event, wind lower but still noteworthy, and no particular remarks of blowing snow reported in the field notes). It can be observed that the case with strong winds: (i) has many more triggered images, (ii) each of those images is much more densely populated by particles (and it is known that number density of blowing snow is higher than in snowfall), (iii) has almost 100% of particles classified as "small particles".

**17. p 12, Fig 8: How is the riming index obtained?**

We added the following clarification in the text, about the riming degree:

 $\ldots$  a continuous riming degree index ranging from 0 to 1, with 1 corresponding to fully developed graupel. The riming degree is a textural information obtained with a supervised classification technique trained on a manually-labelled training set of almost 3400 images, as detailed in Praz et al. (2017).

Please add units and values to the y- axis. All abbreviations used in naming snow particles in the pie-chart must be explained.

The units and values have been added to the histograms/pdfs, and the abbreviations are explained in the caption of the figure.

"The riming index is undefined for "small" particles" - what is the threshold for defining "small" particles?

A threshold is not explicitly defined. These particles have a size that does not allow visual interpretation of their habit and that does not allow computation of geometric descriptors, used in the classification algorithm. Statistically they have a mode of maximum dimension  $(D_{max})$  around 0.45 mm, a median around 0.6 and an interquartile range from 0.4 to 0.8. We clarified the caption as:

The riming index is undefined for "small" particles, i.e. particles that are too small to be identified as a particular hydrometeor class.

18. p. 12, line 53: "...though the curves do not co-fluctuate well". Please specify what do you mean. What is the correlation coefficient?

Our phrasing was misleading. We wanted to draw the attention to the fact that precipitation occurrence is not always synchronized (notably, the evident overestimation of occurrence by ERA-Interim in Summer). This has been rephrased as:

... even though the curves shows some differences in precipitation occurrence.

19. p. 12, line 54: "this optimized Z-S relation provides estimate that are close to the B90A relation.." Specify what is the B90A relation of Matrosov 2009. Why is the local Z-S relation close particularly to B90A? In Table 2, the reference is Matrosov et al 2009, while here Matrosov 2009. Please correct.

The appropriate reference was indeed Matrosov et al. (2009), we thank the reviewer for the correction. The local relation is close in particular to B90A because its parameters are the closest to this relation, among the 6 ones shown. This relation, according to Matrosov et al. (2009), is representative for a various number of mass-to-diameter relationships used in the ice cloud microphysics (it is therefore not representative of a specific snow type).

20. To my opinion, the authors cannot say that they have reduced the uncertainty in quantifying year-long snowfall rates by applying local optimized relationship calculated using only summer measurements. In order to justify this, seasonal evolution of cloud particles has to be analyzed. While this is a subject of future analysis, the authors should discuss limitations of their method.

The concern of the reviewer, that we share, was indeed not underlined enough in the manuscript:

• We removed the only reference to "uncertainty", in page 12, line 53 (original manuscript), rephrasing as:

provides a total cumulated precipitation within the envelope of values of the optimized Z-S relation

• Section 4.3. Rephrasing:

 $\ldots$  while the use of a local Z-S relation, calibrated in the Summer season, allows for a significant reduction of this range of values (from 740 to 989 mm). In this case, however, an important assumption is made, i.e. that this relation can be considered representative for the other seasons.

- In the Summary and Discussion:
  - The MRR estimates were based on a local reflectivity-to-snowfall rate relation, obtained on summer snowfall data only. An important assumption, that will need to be verified or improved, is that we considered this relation as representative for the entire year of MRR measurements.

21. p. 13, fig 9: Caption should be completed and self-explanatory

We rephrased the caption as:

Time series of accumulated snowfall liquid water equivalent. The relations are obtained from Pluvio2 (in blue, for availability periods), censored from phantom precipitation, MRR (in grey the curves corresponding to the relations of Table 2, and ERA-Interim data (in black). Top panel: data corresponding to the year of measurements, from November 2015 to November 2016. Bottom panel: data corresponding to the the Summer campaign 2015/2016, from November 2015 until February 2016.

22. p. 16, line 45 and section 4.2: MRR cannot capture blowing snow... I am asking the authors to clarify this issue and show more clearly using other measurements (Pluvio, AWS) that MRR sees nothing during clear-sky blowing snow events

Because it is known and well documented from past studies that blowing snow is almost never lifted higher than 200 m (first measurement gate of the MRR being 300 m), we wanted to stress that in this configuration (100 m range resolution) the MRR is not the appropriate instrument to monitor blowing snow, and thus we suggest that:

Lower range gate spacing should be employed (lower than 100 m used in the measurements shown here) if blowing snow is of interest.

Unfortunately, the only trusted "ground truth" about blowing snow occurrence is the record of visual observations conducted by Météo France and thus we use them as a reference. We believe that we clarified in the previous points that the instruments suggested by the reviewer cannot be used as a reference for occurrence of clear sky blowing snow events, at least not without further dedicated research.

**Technical corrections**

1. p. 2, line 31: e.g., Konig-Langlo et al.. (as this is not the only paper about human meteorological measurements)

Rephrased, according to the suggestion.

- p. 2, line 37: "for the medium..." delete 'for' Done.
- 3. p. 12 and throughout the text: "cumulated" - $\dot{\delta}$  either accumulated or cumulative Thanks for the suggestion. We changed to "accumulated".
- 4. p. 12, lines 4-5: "within what could be observed...." the whole sentence should be rephrased
- 5. p. 14, Table3: units? mm we?

Indeed. Clarified in the caption, and added to the table.

Figure 1: Inter-comparisons of MRR data collected with two sensors. MRR1, is the MRR used in the originally submitted manuscript (installed within a radome), while MRR2 is a second MRR deployed outside the radome during January/February 2017 (after the analysis period presented in the manuscript).

---

## Author Response (AR2)

**tc-2017-18**

J. Grazioli, *et al*

June 15, 2017

**Responses**

With the present supplementary document we provide our responses to the comments of one anonymous reviewer of the revised manuscript *tc-2017-18*, entitled "Measurements of precipitation in Dumont d'Urville, Terre Adélie, East Antarctica ".

*I would like to thank the authors for addressing all my comments and carefully answering my questions. I have only a few minor comments and some suggestions what could be pointed out more clearly which is in my opinion an important outcome of the study but I leave it to the authors to decide what they want to include.*

We thank the reviewer for the comments, that we address in this document and in the revised version of the manuscript.

- *Abstract, L.4-5: It might be a question from which community you are coming from but in my understanding measurements are from an observing instrument. I cant see how measurements can be based on numerical weather models. I assume that you want to express, that you show here direct observations and not a reanalysis product. But even a reanalysis uses a data assimilation scheme (often from some numerical weather prediction model) to interpolate observations on a model grid but does not provide measurements by itself. I suggest to rephrase this sentence or to leave it out completely. In fact, a satellite person could say that CloudSat already provided precipitation measurements for this region which are not based on numerical weather models as well.*

  We understand the concern of the reviewer, and therefore we decided to rephrase the sentence as follows:

  These instruments collected the first ground-based measurements of precipitation in the region of Terre Adélie (Adélie Land), including precipitation microphysics.

  We underline now that we refer to ground-based (so excluding CloudSat), and direct measurement (excluding numerical weather models).

- *Abstract, L.13-14 and Fig. 9: If our final goal as observationalists is to help modelers to improve their models, I think it would be worth to clearly point out what the biggest problems in terms of precipitation estimates by the models seem to be in that region.*

  We rephrased and clarified the sentence as:

  ERA-Interim overestimates the occurrence of low-intensity precipitation events especially in summer, but it compensates by underestimating the snowfall amounts carried by the most intense events.

  *Your Fig. 9 illustrates that nicely: Even the reanalysis is underestimating the intense snowfall events which seem to be very important for the total amount of accumulated snow. This is in agreement with the findings in the papers by Gorodetskaya et al. If the models would be fixed in a way that they dont produce that unrealistic frequent low intensity snowfall, their underestimation of total snow accumulation would be even worse.*

Fig. 9 is indeed a good visual example, although similar and complementary information is carried also by Fig.11. We decided, following the remark of the reviewer, to better underline in the discussion (before the conclusions) the fact that intense snowfall events are very important at the seasonal/yearly scale, in East Antarctica:

The events of highest intensity, that can contribute to the major part of the yearly snowfall accumulation (e.g. Gorodetskaya et al., 2014, 2015) in East Antarctica, did not occur in summer. This can explain the underestimation of ERA-Interim starting in March 2016.

*I would like to see these aspects more highlighted in the abstract and in the conclusions because I think they are very relevant.*

About the abstract, please refer to the rephrasing proposed just above. In the conclusions, we added the following sentence:

It is worth to underline that the overestimation of occurrence by ERA-Interim partially compensates an observed underestimation of snowfall amounts for the most significant snowfall events. This compensation, over long time periods, may lead to overrate the performance of the model for individual precipitation events.

*It also plays a role for satellite climatologies: If your satellite isnt able to capture the intense snowfall events (e.g. because of infrequent overflies) the derived precip. climatology is very likely biases (underestimates snowfall amount). You could highlight the importance of the different snowfall intensities to the total accumulation (model vs. obs.) by plotting the cumulated total as function of snowfall rate (=snow accum./6h) similar to Fig. 3, right panel in Kulie and Bennartz, JAMC, 2009, Utilizing Spaceborne Radars to Retrieve Dry Snowfall.*

This is a good point. We agree with the reviewer that such a figure will nicely complement Fig.9 and Fig. 11 of the submitted version, and we included it in the revised manuscript (current Fig. 11). We kept, for consistency with Fig. 9 and (previous) Fig. 11, two panels: one for the summer period and one for the entire year. In the text we reference to the figure as:

The overestimation of occurrence compensates the underestimation of the most intense snowfall events, such that at the end of January the total accumulated precipitation of ERA-interim gets close to the one of the Pluvio$^2$. As a result, the contribution of lower snowfall rates to the total accumulated snowfall is much larger for ERA-Interim, with respect to the measurements collected by the MRR and Pluvio$^2$, and this difference is particularly pronounced in the summer period (as shown in Fig. 11)

And:

At 6 h time scale, the yearly snowfall amount is entirely associated to snowfall intensities lower than $2\,\mathrm{mm\,h^{-1}}$ for ERA-Interim, while intensities up to $4.4\,\mathrm{mm\,h^{-1}}$ have been measured with the MRR. (Fig. 11)

**Typos:**

1. *P.2, L. 11: reanalysis of, probably better reanalysis using or reanalysis based on*
   Corrected, according to the suggestion of the reviewer.

2. *P.17, L.11: In-text citation instead of (Palm et al.)*
   Amended.

3. P.21, L.8: We thanks
   The typo has been corrected.

4. P.21m L.14: poof reading
   The typo has been corrected.

**References**

[revised manuscript text omitted]